# Characterization of nucleolar SUMO isopeptidases unveils a general p53-independent checkpoint of impaired ribosome biogenesis

Judith Dönig[1], Hannah Mende[1], Jimena Davila Gallesio[2], Kristina Wagner[1], Paul Hotz [1], Kathrin Schunck[1,7], Tanja Piller [1,8], Soraya Hölper[1,8], Sara Uhan[3,4,5], Manuel Kaulich [1], Matthias Wirth [3,4,5,6], Ulrich Keller [3,4,5], Georg Tascher [1], Katherine E. Bohnsack [2] & Stefan Müller [1,4] ✉

Ribosome biogenesis is a multi-step process, in which a network of *trans*-acting factors ensures the coordinated assembly of pre-ribosomal particles in order to generate functional ribosomes. Ribosome biogenesis is tightly coordinated with cell proliferation and its perturbation activates a p53-dependent cell-cycle checkpoint. How p53-independent signalling networks connect impaired ribosome biogenesis to the cell-cycle machinery has remained largely enigmatic. We demonstrate that inactivation of the nucleolar SUMO isopeptidases SENP3 and SENP5 disturbs distinct steps of 40S and 60S ribosomal subunit assembly pathways, thereby triggering the canonical p53-dependent impaired ribosome biogenesis checkpoint. However, inactivation of SENP3 or SENP5 also induces a p53-independent checkpoint that converges on the specific downregulation of the key cell-cycle regulator CDK6. We further reveal that impaired ribosome biogenesis generally triggers the downregulation of CDK6, independent of the cellular p53 status. Altogether, these data define the role of SUMO signalling in ribosome biogenesis and unveil a p53-independent checkpoint of impaired ribosome biogenesis.

Ribosomes are the key molecular machinery responsible for cellular protein synthesis. The eukaryotic cytosolic 80S ribosome is composed of a small 40S and a large 60S subunit. In humans, the 60S subunit comprises 47 ribosomal proteins (RPs) and three rRNA species (28S, 5.8S, and 5S), the 40S subunit is composed of 33 RPs and the 18S rRNA. The 28S, 18S, and 5.8S rRNAs are derived from serial processing of a polycistronic precursor RNA that is transcribed in the nucleolus by RNA polymerase I[1]. Ribosome biogenesis, i.e., processing and assembly of rRNAs with ribosomal proteins, is orchestrated by a network of over 250 non-ribosomal proteins, known as *trans*-acting factors[2,3]. Early biogenesis factors and ribosomal proteins associate with the nascent pre-rRNA to form the first stable pre-ribosomal particle known as the

[1]Institute of Biochemistry II, Goethe University Frankfurt, Medical Faculty, Theodor-Stern-Kai 7, 60590 Frankfurt, Germany. [2]Department of Molecular Biology, University Medical Centre Göttingen, Humboldtallee 23, 37073 Göttingen, Germany. [3]Department of Hematology, Oncology and Cancer Immunology (Campus Benjamin Franklin), Charité Universitätsmedizin Berlin, corporate member of Freie Universität Berlin and Humboldt-Universität zu Berlin, Hindenburgdamm 30, 12203 Berlin, Germany. [4]German Cancer Consortium (DKTK), Im Neuenheimer Feld 280, 69120 Heidelberg, Germany. [5]Max Delbrück Center, Robert-Rössle-Str. 10, 13125 Berlin, Germany. [6]Department of General, Visceral and Pediatric Surgery, University Medical Center Göttingen, Robert-Koch-Str. 40, 37075 Göttingen, Germany. [7]Present address: PharmBioTec gGmbH, Schiffweiler, Germany. [8]Present address: Sanofi AG, Frankfurt, Germany. ✉e-mail: ste.mueller@em.uni-frankfurt.de

90S pre-ribosome or small subunit (SSU) processome[4]. Dismantling the 90S particle generates pre-40S and pre-60S particles that follow separate maturation pathways leading ultimately to the production of translation-competent 80S ribosomes.

Previous work from our group and others demonstrated that the ubiquitin-like SUMO system is involved in ribosome maturation[5–8]. SUMOylation is a ubiquitin-related pathway where the modifier SUMO (SUMO1 or the highly related SUMO2/3 proteins) is covalently attached to lysine residues of target proteins in a multistep enzymatic process involving an E1 activating enzyme, an E2 conjugating enzyme and E3 SUMO ligases[9]. SUMOylation typically controls the dynamics of protein assemblies through either preventing or promoting distinct protein-protein interactions[10]. SUMOylation can be reversed by SUMO deconjugases of the ULP/SENP family, whose members share conserved catalytic domains. Human cells express six SENP family members, which exhibit characteristic subcellular distributions[11]. Notably, the SENP family members SENP3 and SENP5 are compartmentalized within the granular component of the nucleolus[12,13], a sub-nucleolar compartment where distinct pre-rRNA processing steps take place. Genetic experiments in the yeast *Saccharomyces cerevisiae* initially demonstrated that mutations in the SUMO conjugation enzyme UBC9 as well as the deconjugase ULP1 cause pre-60S export defects[6]. In mammalian cells, we showed that the ULP1-related SENP3 protease is involved in pre-60S maturation by controlling SUMOylation processes at 60S pre-ribosomes[6]. We identified the pre-60S maturation factor PELP1 as a target of SENP3 and found that balanced SUMO conjugation-deconjugation of PELP1 controls the timely recruitment and release of the ribosome re-modeling factor MDN1, thereby influencing 28S rRNA maturation[14,15]. While this positions SENP3 in the pre-60S maturation process, the function and substrate specificity of SENP5 have remained largely enigmatic.

Ribosome biogenesis is an energy-demanding process that requires tight coordination with cell growth and proliferation[16]. Highly proliferating cells typically show enhanced rates of ribosome biogenesis to enable them to meet the needs for extensive and efficient protein synthesis. This is best exemplified in tumor cells, where the MYC oncoprotein often functions as a common driver of cell cycle progression and ribosome biogenesis[17]. By contrast, impaired ribosome biogenesis initiates signaling cascades that block cell cycle progression. The best-understood pathway connecting disturbed ribosome biogenesis with cell cycle progression is the p53-mediated impaired ribosome biogenesis checkpoint (IRBC)[18–20]. This pathway is triggered by the accumulation of distinct unassembled RPs, in particular RPL5 and RPL11, which form a complex with the 5S rRNA (5S RNP)[21]. This complex sequesters the E3 ubiquitin ligase MDM2 thereby promoting the stabilization of its target p53.

Here, we undertook an unbiased proteomic profiling approach to define the landscape of SENP3 and SENP5 functions. We provide evidence that both SUMO deconjugases are involved in ribosomal subunit maturation and further demonstrate that their inactivation affects cell cycle progression in both p53-proficient and deficient cells. Impaired cell cycle progression upon SENP3 or SENP5 depletion was associated with downregulation of CDK6. We discovered that downregulation of CDK6 is a general response to perturbed ribosome biogenesis, revealing a so-far unknown p53-independent signaling axis that connects impaired ribosome biogenesis to cell cycle progression.

## Results

### A comparative network of SENP3 and SENP5-controlled SUMOylation

To unravel the target specificities and distinct functions of SENP3 and SENP5, we aimed to develop a comparative network of their respective interaction partners and targets. To this end, we used two complementary mass spectrometry (MS)-based approaches. We performed interactomics by analyzing proteins associated with catalytically inactive, FLAG-tagged SENP3 or SENP5 expressed in HEK293T cells. Expression and correct nucleolar localization of the constructs was validated by anti-FLAG immunoblotting and immunofluorescence (Supplementary Fig. 1a, b). Furthermore, we used anti-SUMO2 affinity purification followed by MS in control cells and cells depleted of each isopeptidase to identify potential target proteins that exhibit enhanced modification (Supplementary Fig. 1c). SUMO2/3 affinity enrichment was done based on the preferential activity of SENP3 and SENP5 towards SUMO2/3 conjugates. By FLAG-IP-MS (immunoprecipitation followed by mass spectrometry), we enriched 124 SENP3-associated proteins (Fig. 1a, c and Supplementary Data 1), while comparative proteomics after anti-SUMO2 affinity purification identified 63 proteins that exhibited at least 1.5-fold enhanced SUMOylation in cells lacking SENP3 (Fig. 1b, c and Supplementary Data 2). These changes are not simply due to alterations in protein expression as confirmed by proteome analysis of control cells and cells depleted from SENP3 (Supplementary Data 3). GO term analysis of interactors and potential targets revealed a significant functional enrichment of proteins involved in pre-rRNA processing and ribosome biogenesis (Fig. 1d). Accordingly, STRING network analysis shows a highly interconnected network of ribosomal proteins and ribosome biogenesis factors (Fig. 1e). The two central nodes cluster around components of the SSU processome/pre-40S ribosome and the mammalian rixosome, which is defined by PELP1. The term rixosome refers to Rix1, the yeast orthologue of PELP1. The mammalian rixosome comprises the core components PELP1, WDR18, TEX10 and the associated factors MDN1, LAS1L and NOL9[22]. We and others previously identified the mammalian rixosome as a regulator of pre-60S maturation and identified PELP1 and LAS1L as prime functional targets of SENP3[14]. The new dataset now identifies the rixosome core components PELP1 and TEX10 as well as the associated factors, LAS1L, NOL9 and MDN1 as either interactors or potential targets of SENP3 (Fig. 1a–c, e). In fact, LAS1L, TEX10 and PELP1 were present in both datasets demonstrating that our unbiased, two-pillar proteomics strategy enables the reliable identification of bona fide SENP-controlled pathways (Fig. 1c).

We therefore used this approach to next profile the target specificity of SENP5, which was largely uncharacterized. We identified 109 SENP5-associated proteins and 51 proteins, whose SUMOylation was at least 1.5-fold enhanced upon loss of SENP5 (Fig. 2a–c and Supplementary Data 1 and 4). Again, changes are not due to alterations in protein expression (Supplementary Data 5). Similar to what was observed for SENP3, GO term analysis of SENP5 interactors and targets revealed ribosome biogenesis as the most significantly enriched biological process (Fig. 2d). Notably, however, STRING network analysis shows that SENP5 is not linked to the rixosome complex, but—like SENP3—is connected to the SSU processome/pre-40S node (Fig. 2e). Within this node UTP14A, which is involved in pre-40S maturation, exhibits by far the most strongly upregulated SUMOylation upon depletion of SENP5 (Fig. 2b). UTP14A was also found to be physically associated with SENP3 (Supplementary Data 1) defining UTP14A as a potential key target of SENP3 and/or SENP5 in the 40S assembly process. By co-immunoprecipitation, we validated binding of both SENP3 and SENP5 to UTP14A (Fig. 2f). Furthermore, upon individual depletion of either SENP3 or SENP5 or co-depletion of both proteins, anti-UTP14A immunoblotting of whole-cell extracts detected UTP14A at its expected apparent MW of 100 kDa, but upon longer exposure revealed an additional higher-molecular weight species migrating at 140 kDa, a size difference that is consistent with a SUMO conjugated from of UTP14A (Fig. 2g, left panel). In line with this interpretation, the 140 kDa species was absent when cells were treated with the highly specific SUMO E1 inhibitor (SUMOi) TAK-981 (Supplementary Fig. 1d, left panel)[23]. Furthermore, the 140 kDa species was detected by anti-UTP14A immunoblotting in anti-SUMO2/3 immunoprecipitates (Fig. 2g, right panel and Supplementary Fig. 1e). Individual depletion of SENP3 or SENP5 enhances SUMOylation of endogenous UTP14A, with

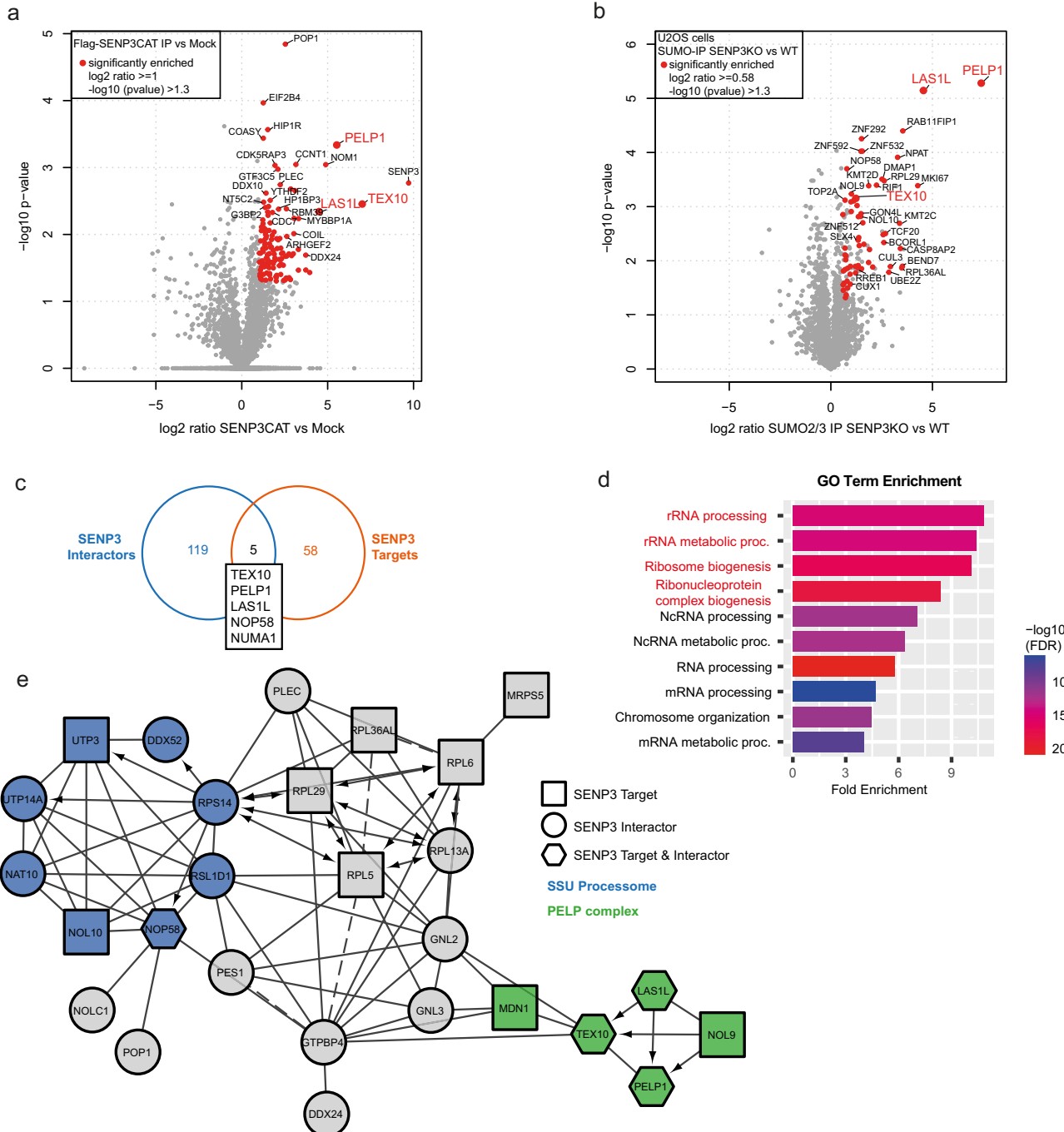

**Fig. 1 | The network of SENP3 interactors and targets. a** Volcano plot of quantitative MS analysis after anti-FLAG-IP of HEK923T cell lysates transiently expressing FLAG-SENP3$^{C532S}$. Significantly enriched interactors are color-coded (log2 ratio ≥ 1, −log10 $p$ value ≥ 1.3). Identification of candidates is based on two-sided Student's $t$-test analysis comparing LFQ intensities of IPs from FLAG-SENP3 expressing and control cells. Members of the PELP1 complex are additionally highlighted. Experiments were performed in triplicates. **b** Volcano plot summarizing the results of a quantitative MS analysis comparing SUMO targets of U2OS WT and U2OS SENP3 KO cells. Proteins were enriched by anti-SUMO2/3 IP. Hits considered as significant are highlighted in red (log2 ratio ≥ 0.58, −log10 $p$ value ≥ 1.3). Candidate SENP3 target proteins were defined by two-sided Student's $t$-test analysis comparing LFQ intensities of anti-SUMO2/3 IP with the respective IgG control IPs. Components of the PELP1 complex are additionally highlighted. Experiments were performed as triplicates. **c** Venn diagram representing the overlap of proteins identified as SENP3 interactors (**a**) or SENP3 targets (**b**). **d** Gene Ontology term enrichment analysis of biological processes (GO BP) of high-confidence SENP3 interactors (**a**) and targets (**b**) identified by MS (interactors: log2 ratio ≥ 1, targets: log2 ratio ≥ 0.58). Shown here are the top 10 enriched biological processes. Highlighted in red are terms directly connected to the process of ribosome biogenesis. The enrichment analysis was done using the ShinyGO tool, applying an FDR cutoff of 0.05. **e** The interconnection of significantly enriched SENP3 interactors and targets (**a**, **b**) visualized as STRING network. The minimum required interaction score was set to high confidence (0.7). Proteins that are part of the PELP1 complex are color-coded in green, those being part of the SSU processome are marked in blue. Only connected proteins are visualized.

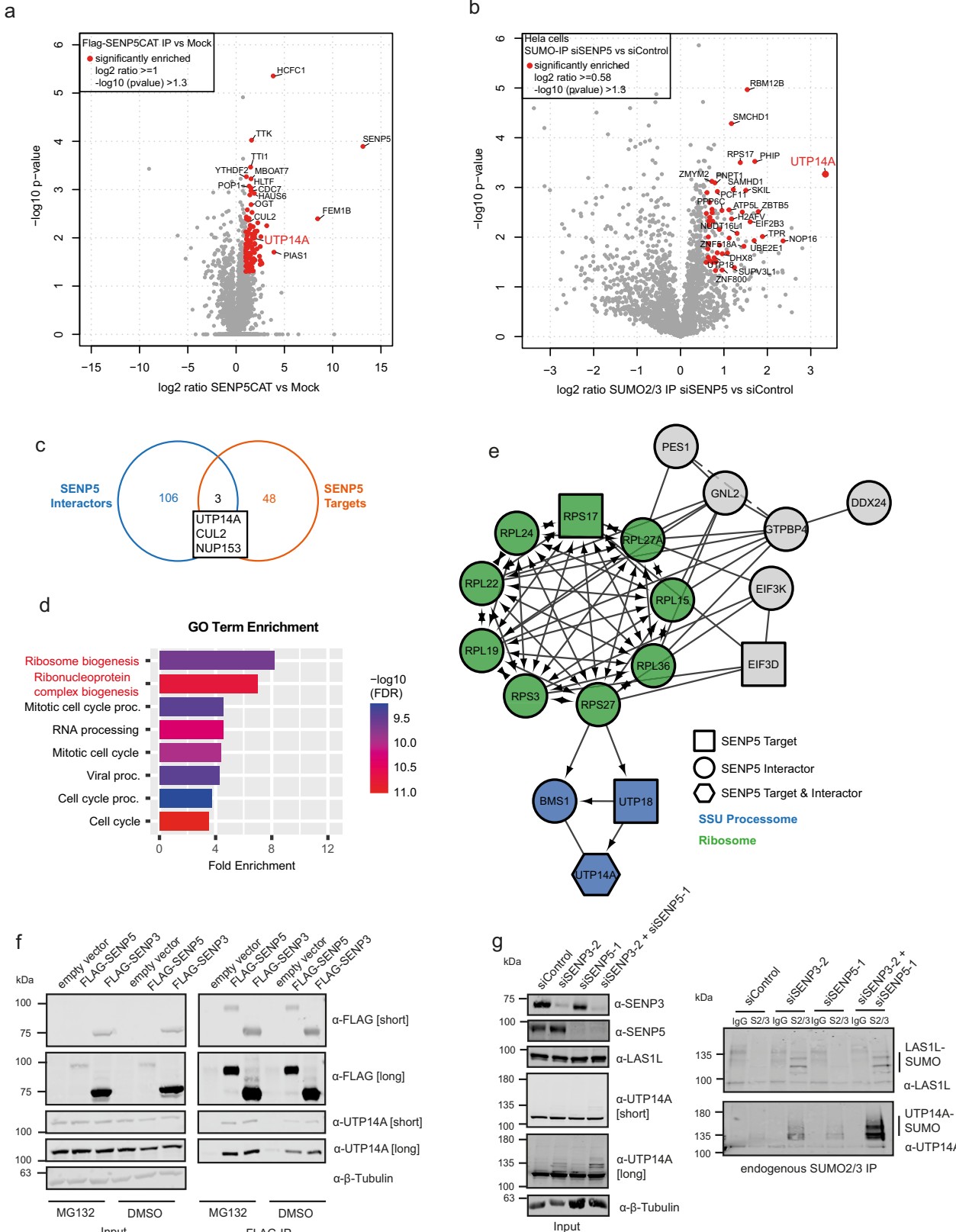

SENP3 depletion exhibiting more pronounced effects. Notably, co-depletion of both proteins further increases the level of UTP14A-SUMO2/3 (Fig. 2g, right panel). To address whether SENP3/5 regulate other pre-40S components, we tested UTP18 and BMS1. Both proteins exhibit enhanced SUMO2/3 conjugation upon depletion of SENP3 and further enhancement upon co-depletion of SENP5, while individual depletion of SENP5 only marginally affects the modification (Supplementary Fig. 1e). These data indicate that UTP14A is indeed a potential prime substrate of SENP3/5 and further suggest that both iso-peptidases exert partially redundant functions as deconjugases of UTP14A and potentially other pre-40S regulators. This contrasts with the activities of SENP3 and SENP5 towards the rixosome complex,

**Fig. 2 | The network of SENP5 interactors and targets. a** Volcano plot of quantitative MS analysis after anti-FLAG-IP of HEK923T cell lysates transiently expressing FLAG-SENP5[C713S]. Significantly enriched interactors are color-coded (log2 ratio ≥ 1, −log10 p value ≥ 1.3). Identification of candidates is based on two-sided Student's t-test analysis comparing LFQ intensities of IPs from FLAG-SENP5 expressing and control cells. Experiments were performed as triplicates. **b** Volcano plot summarizing the results of a quantitative MS analysis comparing SUMO targets of HeLa cells transfected with siSENP5 or siControl. Proteins were enriched by anti-SUMO2/3 IP. Hits considered as significant are highlighted in red (log2 ratio ≥ 0.58, −log10 p value ≥ 1.3). Candidate SENP5 target proteins were defined by two-sided Student's t-test analysis comparing LFQ intensities of anti-SUMO2/3 IP with the respective IgG control IPs. The experiment was performed with four replicates. **c** Venn diagram representing the overlap of proteins identified as SENP5 interactors (**a**) or SENP5 targets (**b**). **d** High-confidence SENP5 targets and interactors were subjected to GO BP analysis (interactors: log2 ratio ≥ 1, targets: log2 ratio ≥ 0.58). Displayed here are

8 out of the top 10 enriched processes. Highlighted in red are terms connected to ribosome biogenesis. The enrichment analysis was done using the ShinyGO tool, applying an FDR cutoff of 0.05. **e** STRING network analysis of significantly enriched interactors and targets of SENP5 (**a, b**). Ribosomal proteins or components of the SSU processome are highlighted in green or blue, respectively. Only connected proteins are visualized. **f** HEK293T cells were transfected with FLAG-tagged SENP3, SENP5 or empty vector control for 48 h. To stabilize short-lived SENP5, cells were treated with MG-132 (25 μM) or DMSO for 4 h prior to IP. Immunoblotting was performed as indicated. Input samples are shown in the left panel. Probing against the FLAG-tag was used to control the expression of the FLAG-tagged SENP3 or SENP5 constructs. **g** HeLa cells were transfected with siRNAs against SENP3, SENP5, SENP3 + SENP5 or siControl. 72 h post transfection, endogenous SUMO2/3 or IgG control IP were performed and samples were analyzed by immunoblotting as indicated. Input samples are shown in the left panel. Source data for (**f**) and (**g**) are provided as a Source Data file.

where loss of SENP5 alone or in combination with SENP3 does not affect the modification of LAS1L indicating that SENP5 does not catalyze deSUMOylation at pre-60S particles (Fig. 2g, right panel). Altogether, our datasets strengthen the notion that SENP3 and SENP5 are involved in ribosome biogenesis by controlling the SUMOylation status of distinct trans-acting factors.

## Nucleolar SENPs control the association of UTP14A with pre-40/SSU components

Based on the above findings, we aimed to determine how SENP3/5 control UTP14A functions. We therefore investigated whether SENP3/5 regulate the association of UTP14A with distinct interaction partners. To this end, we immunoprecipitated endogenous UTP14A from control cells or cells depleted from SENP3, SENP5 or SENP3/5 and analyzed immunoprecipitated proteins by MS (Supplementary Data 6). In control cells, 29 proteins were at least 2-fold enriched in UTP14A IPs compared to IgG controls (Fig. 3a). STRING and GO term analysis of these proteins reveals a strong enrichment of an interconnected network of RPS proteins and 40S biogenesis factors (Fig. 3b, c). Upon individual depletion of either SENP3 or SENP5 or co-depletion of both proteins, a set of RPS proteins and 40S trans-acting factors, including RRP12, TSR1 and NOP14, exhibit reduced association of UTP14A (Fig. 3d and Supplementary Data 6). Proteome analysis confirmed that this was not due to alterations in the expression of the respective proteins (Supplementary Data 7). We therefore hypothesized that the association of UTP14A with pre-40S particles is impaired upon depletion of SENP3/5. To validate this point, we performed sucrose gradient sedimentation assays following depletion of SENP3/5. In control cells, UTP14A co-fractionates with RPS3A in pre-40S particles (fraction 5-8) (Fig. 3e). However, upon depletion of SENP3/5 the amount of UTP14A is reduced in these fractions supporting the idea that unrestricted SUMOylation impairs the association of UTP14A with pre-40S/SSU processome particles (Supplementary Fig. 2a). In agreement with these data, UTP14A is shifted from the nucleolus to the nucleoplasmic fraction in the absence of either SENP3 or SENP5 (Supplementary Fig. 2b, c). Taking advantage of published MS datasets, which identified K733 as the major SUMOylation site in UTP14A[24], we confirmed that K733 is also the major SUMO site targeted by SENP3/5 (Supplementary Fig. 2d). Based on the available cryo-EM structures we therefore hypothesize that in case of UTP14A, the attachment of SUMO at K733 sterically hinders the incorporation of UTP14A in pre-ribosomes[25] (Supplementary Fig. 2e). Notably, in contrast to UTP14A, the association of PELP1 with pre-60S particles was enhanced upon SENP3/5 depletion supporting our previous concept of rixosome trapping at pre-60S particles upon constitutive SUMOylation. Altogether, these data indicate that dynamics of SUMO conjugation-deconjugation shape pre-ribosomal particles at specific stages of the maturation pathway.

## Depletion of nucleolar SENPs impairs pre-rRNA processing and ribosome biogenesis

The above data suggest that SENP3 is connected to both pre-40S and pre-60S maturation steps, while SENP5 is predominately linked to the 40S assembly pathway. In support of this view, sucrose gradient sedimentation assays revealed that endogenous SENP3 co-fractionates with both pre-40S and pre-60S particles, while SENP5 more prominently co-sediments with pre-40S ribosomes (Fig. 4a). To more directly explore whether SENP3 and SENP5 are required for efficient pre-rRNA processing, we performed northern blots using specific probes hybridizing within the internal transcribed spacers ITS1 and ITS2 that detect all major pre-rRNA processing intermediates (Supplementary Fig. 3a). In accordance with our published findings, depletion of SENP3 results in the accumulation of the 32S precursor demonstrating the involvement of SENP3 in conversion of this pre-rRNA to the mature 28S rRNA of the 60S subunit (Fig. 4b and Supplementary Fig. 3a). In line with the findings that SENP5 does not interact with or target the rixosome components or other pre-60S biogenesis factors, depletion of SENP5 does not affect 32S levels. By contrast, depletion of either SENP5 or SENP3 reduces the level of the 18SE rRNA species (Fig. 4b and Supplementary Fig. 3a), which represents a direct precursor of the 18S rRNA of the 40S subunit, indicating that both nucleolar isopeptidases are important for small subunit maturation. To monitor the impact of SENP3 and SENP5 on the maturation of the small and large ribosomal subunits, we performed Ribo-Halo assays by using genetically encoded reporters of both subunits[26]. Using CRISPR/Cas-based genome engineering, we fused a Halo cassette to the C-terminus of the endogenous *RPL28 (=eL28)* or *RPS3 (=uS3)* genes generating Halo-tagged RPL28/RPS3 expressing HeLa cell lines (Supplementary Fig. 3b). Structural modeling as well as sucrose gradient sedimentation assays of RPL28-Halo and RPS3-Halo cells support proper integration of both fusion proteins into mature ribosomes, also validating published data for RPS3-Halo[26] (Supplementary Fig. 3c, d). Cells expressing the Halo-fused proteins were sequentially labeled with two distinct fluorescent ligands to follow the fate of existing and newly synthesized 40S or 60S ribosomal subunits by immunofluorescence (Fig. 4c). To monitor "old" (pre-existing) or nascent 40S/60S ribosomal subunits, we performed sequential labeling with individual red (tetramethylrhodamine; TMR) or green (R110) fluorescent ligands. Imaging of the control RPL28-Halo cells showed a predominant yellow signal resulting from a merge of both "old/red" and newly synthesized/green ribosomal subunits (Fig. 4d). By contrast, upon siRNA-mediated depletion of the pre-60S maturation factor MDN1, the red signal of the old ribosomes was largely dominating in the cytosol, indicating that synthesis of 60S ribosomal subunit was impaired (Fig. 4d, e). A very similar phenotype was observed upon depletion of SENP3, confirming that SENP3 is indeed required for proper 60S maturation (Fig. 4d, e). Consistent

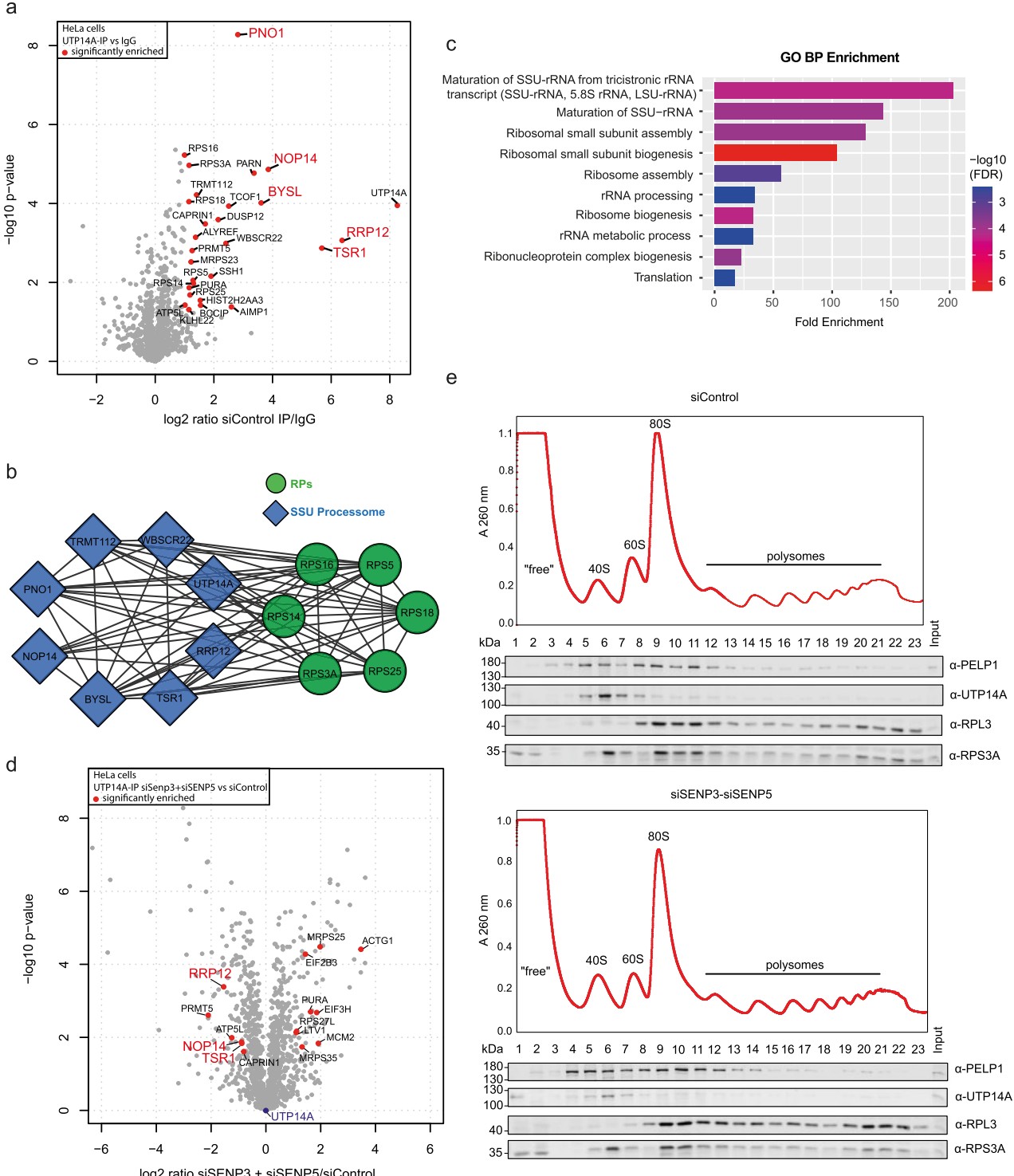

with the data from northern blotting analysis of the pre-rRNA processing, depletion of SENP5 did not impair 60S biogenesis (Fig. 4d, e).

To see how SENP3/5 affect pre-40S maturation, analogous experiments were performed in the RPS3-Halo cell line. Control cells again showed the presence of both "old" and newly synthesized 40S subunits as indicated by the pre-dominantly yellow (merge "old" and "new" ribosomes) signal in the cytosol (Fig. 4d). Depletion of UTP14A, which was used as a control, shifts the balance towards a red signal indicative of impaired de novo synthesis of 40S subunits (Fig. 4d, e). A

comparable scenario was detected when either SENP3 or SENP5 were individually depleted or co-depleted from RPS3-Halo cells (Fig. 4d, e and Supplementary Fig. 3e). Quantification of the signals demonstrates that lack of SENP3 has a stronger impact on 40S biogenesis than the lack of SENP5 (Fig. 4e). Importantly, however, co-depletion of both isopeptidases further aggravates the maturation defect (Supplementary Fig. 3e). Altogether, these data demonstrate that nucleolar SENPs are important regulators of ribosome biogenesis with SENP3 controlling both 40 and 60S maturation and SENP5 exhibiting a more specific role in 40S maturation.

**Fig. 3 | SENP3 and SENP5 deficiency affects the UTP14A interactome and its pre-40S association. a** Volcano plot of UTP14A interactors identified by quantitative MS. HeLa cells were transfected with control siRNA for 72 h and endogenous UTP14A or control IP was performed. Significantly enriched proteins are indicated by red dots (log2 ratio ≥ 1, −log10 p value ≥ 1.3). Two-sided Student's t-test analysis was performed to identify UTP14A interactors by comparing LFQ intensities of UTP14A-IP with IgG control IPs. **b** Significantly enriched UTP14A interactors identified in (**a**) are represented as STRING networks. Only high confidence (minimum interaction score 0.7) connections are shown. rProteins are highlighted in green, components of the SSU processome are marked in blue. Only connected proteins are visualized. **c** GO BP analysis of identified UPT14A interactors (highlighted in **a**). An FDR cutoff of 0.05 was applied and only the top 10 enriched biological processes are shown here. The ShinyGO online tool was used for analysis. **d** UTP14A interactome upon 72 h of siSENP3 and siSENP5 KD visualized as volcano plot. Two-sided Student's t-test was performed. Significantly upregulated hits (log2 ratio ≥ 1 in siSENP3/5 UTP14A-IP vs siSENP3/5 IgG IP and log2 ratio ≥ 0.58 in siSENP3/5 UTP14A-IP vs siControl UTP14A-IP) and significantly downregulated hits (log2 ratio ≥ 1 in siControl UTP14A-IP vs siControl IgG IP and log2 ratio ≤ −0.58 in siSENP3/5 UTP14A-IP vs siControl UTP14A-IP) are highlighted in red. **e** Lysates of cycloheximide-treated HEK293T cells depleted of SENP3/5 or transfected with control siRNA were subjected to sucrose gradient density centrifugation. Top: respective absorbance profiles at 260 nm. Bottom: immunoblots to monitor the presence of indicated proteins in each fraction. Experiments were performed with three replicates. Source data for (**e**) are provided as a Source Data file.

## Depletion of nucleolar SENPs triggers the canonical p53-dependent impaired ribosome biogenesis checkpoint

To unravel how lack of SENP3 or SENP5 affect cellular signaling pathways, we performed TMT-based quantitative proteomics and RNA-Seq in U2OS cells following depletion of SENP3 with two specific siRNAs (Supplementary Fig. 4a). Comparison with control cells revealed a common set of 199 proteins that were either up- or downregulated with both siRNAs at least 1.5-fold (Supplementary Data 8). Among these, 85 proteins were downregulated and 114 proteins were upregulated (Fig. 5a and Supplementary Fig. 4b, c). Importantly, functional clustering of the altered proteins by KEGG pathway analysis revealed a typical p53 signature (Fig. 5b). In the RNA-Seq transcriptomic analysis, differentially expressed transcripts were filtered using a fold change of >2. Here, we identified 386 upregulated and 616 downregulated transcripts upon RNAi-mediated depletion by both siRNAs (Supplementary Data 9). Gene set enrichment analysis (GSEA) of this dataset showed the p53 pathway as the top significantly enriched hallmark gene set (Fig. 5c, d). We validated the upregulation of p53 as well as its key target p21 (CDKN1A), by immunoblotting in whole-cell extracts from SENP3-depleted U2OS cells (Fig. 5e, f). Similarly, depletion of SENP5 with two independent siRNAs also triggered induction of p21 (Fig. 5f). Consistent with activated p53 signaling, loss of SENP3 or SENP5 altered the cell cycle profile by decreasing the proportion of S phase cells (Fig. 5g, h). Accordingly, the fraction of cells in G1 or G2 was increased. Altogether, these results are indicative of a typical p53-induced G1/S and G2/M arrest. When compared to siSENP3-2, induction of p21 was more pronounced in siSENP3-1 depleted cells, likely explaining the observed G2/M block[27]. Depletion of SENP5 also reduced the fraction of S phase cells by about 50%, which was accompanied by an increased number of cells in G1 (Fig. 5h). Taken together, these data indicate that nucleolar SENPs are required for proper ribosome biogenesis and accordingly, that their depletion triggers the canonical p53-dependent impaired ribosome biogenesis checkpoint.

## SENP3 and SENP5 deficiency induces loss of CDK6 and impairment of cell cycle progression in a p53-independent process

To unveil p53-independent signaling events that are affected by SENP3 or SENP5 deficiency, we depleted SENP3 from SAOS-2 cells, which lack functional p53 and analyzed the proteome and transcriptome by TMT-based quantitative proteomics and RNA-Seq (Supplementary Fig. 5a and Supplementary Data 10 and 11). Comparison with control cells revealed a common set of 50 proteins that were at least 1.5-fold up- or downregulated with the two unrelated SENP3 siRNAs (Supplementary Data 10). Among these, 30 proteins were downregulated and 20 proteins were upregulated (Fig. 6a and Supplementary Fig. 5b, c). Analysis of the MS data identified CDK6 among the most strongly downregulated proteins with both siRNAs. Comparison with the proteome dataset from SENP3-depleted U2OS cells also revealed a strong downregulation of CDK6 in U2OS cells (Fig. 5a). RNA-Seq data from SENP3-depleted cells similarly show reduced levels of CDK6 transcripts in both U2OS and SAOS-2 cells (Supplementary Fig. 5d and Supplementary Data 11). Importantly, when merging all proteomics and RNA-

Seq datasets derived from SENP3-depleted U2OS or SAOS-2 cells, CDK6 represents the only common downregulated target in all experiments (Fig. 6b). We validated the downregulation of CDK6 by immunoblotting of cell extracts from control cells or SENP3-depleted U2OS or SAOS-2 cells (Fig. 6c). To rule out off-target effects we used four independent siRNAs directed against SENP3 and included two unrelated control siRNAs, one of them targeting SENP6 (Fig. 6d). RT-qPCR confirmed the reduced CDK6 mRNA level upon depletion of SENP3, indicating that CDK6 loss upon SENP3 deficiency is likely due to transcriptional or post-transcriptional regulation, rather than post-translational processes, such as ubiquitin-proteasome-mediated degradation (Fig. 6e). In agreement with this, treatment of cells with the proteasome inhibitor MG-132 did not restore CDK6 protein levels in the absence of SENP3 (Supplementary Fig. 5e). Importantly, depletion of SENP5 also diminished CDK6 expression (Fig. 6f), albeit to a slightly lesser extent. Upon association with CyclinD family members, CDK6 primarily drives G1 progression and G1/S transition. In most cell types, CDK6 acts in conjunction with the related CDK4 protein. Notably, however, CDK4 levels were not affected by either depletion of SENP3 or SENP5, pointing to a specific role of both isopeptidases in CDK6-mediated cell cycle control (Fig. 6f and Supplementary Fig. 5f).

We therefore asked whether depletion of SENP3/5 can affect cell cycle progression in a p53-independent pathway. To provide a p53-proficient and p53-deficient cellular system in an isogenic background, we used wild-type RPE1 cells or RPE1$^{\Delta p53}$ cells, in which p53 was genetically inactivated by CRISPR-Cas-mediated gene inactivation. RPE1 is a diploid, non-transformed, hTERT-immortalized cell line derived from retinal pigment epithelial cells. In this cellular setting, we confirmed that loss of SENP3 downregulates CDK6, irrespective of the p53 status (Fig. 6g). Furthermore, we validated that depletion of CDK6 disturbs cell cycle progression and diminishes the S phase population in both parental RPE1 and RPE1$^{\Delta p53}$ cells, indicating that CDK6 is involved in G1/S progression in RPE1 cells (Supplementary Fig. 6a). Importantly, the absence of either SENP3 or SENP5 also hampers cell cycle progression in both p53-proficient and p53-deficient cells. Depletion of SENP3 in parental RPE1 (WT p53) cells strongly reduced the fraction of S phase cells, suggesting that activation of a p53 response upon SENP3 loss limits G1/S transition in RPE1 cells. Notably, we also observed an increase in the number of cells in G2, indicating that a fraction of RPE1 cells also underwent a G/2M arrest (Fig. 7a, left panel). Most importantly, depletion of SENP3 in RPE1$^{\Delta p53}$ cells also reduced the fraction of cells in S phase and concomitantly increased the G1 population thereby largely phenocopying the effects observed upon depletion of CDK6 (Fig. 7a, right panel and Supplementary Fig. 6a). Consistent with these findings, depletion of SENP3 slowed cell proliferation of RPE1$^{WT}$ and RPE1$^{\Delta p53}$ cells (Supplementary Fig. 6b). We next aimed to monitor whether and how depletion of SENP5 affects the cell cycle profile in p53-proficient or deficient RPE1 cells. Similar to what we observed upon depletion of SENP3, the number of S phase cells was also strongly decreased upon depletion of SENP5 in both the parental RPE1 and the RPE1$^{\Delta p53}$ cells (Fig. 7b). Moreover, co-depletion of both isopeptidases reduces the number of S

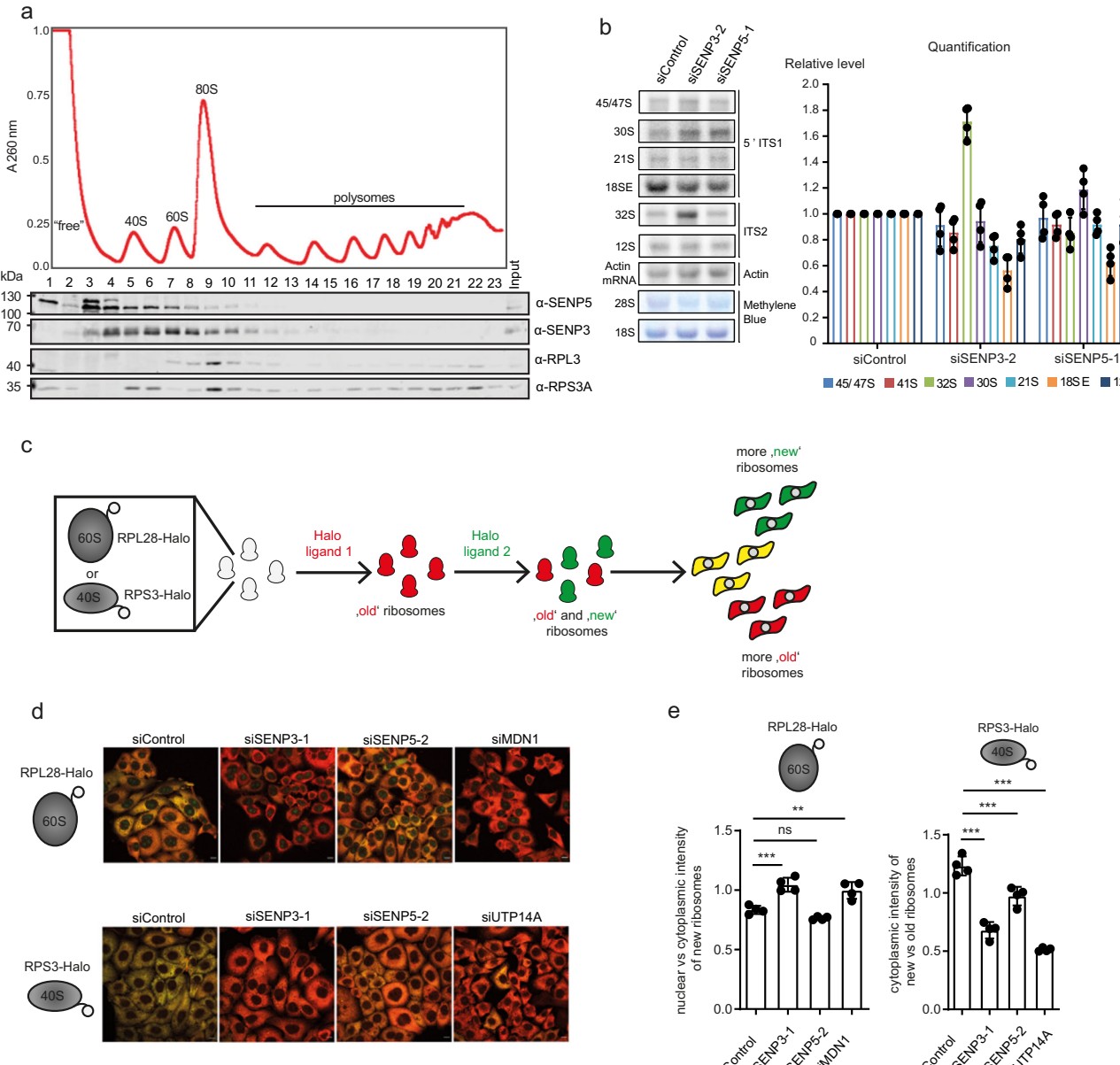

**Fig. 4 | SENP3 and SENP5 deficiency leads to defects in ribosome maturation.**
**a** Whole-cell lysates from cycloheximide-treated HEK293T cells were separated using sucrose density centrifugation. Peaks corresponding to (pre-)ribosomal complexes are marked on an absorbance profile at 260 nm (top). The presence of the indicated proteins in each fraction of the gradient was determined by western blotting. Experiments were performed with two replicates. **b** Total RNAs extracted from siRNA-treated cells were separated by denaturing agarose gel electrophoresis and transferred to nylon membrane. 28S and 18S rRNAs were detected by methylene blue staining. Pre-rRNA species were detected using [$^{32}$P]-labeled probe hybridizing in the internal transcribed spacers 1 (5′ ITS1) and 2 (ITS2). The actin mRNA served as a loading control. The levels of major pre-rRNA species were quantified in four experiments and are shown as mean ± standard deviation. Statistical significance was determined using the two-tailed Student's *t*-test. **c** Simplified workflow of the Ribo-Halo assay used to monitor the cellular

production of either 60S or 40S ribosomal subunits. RPL28 or RPS3 were endogenously tagged with the HaloTag, using the CRISPR/Cas system. Afterward, cells were treated with two different Halo ligands (R110-ligand, TMR ligand) to finally follow the production and export of both ribosomal subunits with fluorescence microscopy. **d** Representative fluorescent images of HeLa RPL28-Halo (top) and RPS3-Halo (bottom) cells transfected as indicated and stained with Halo ligands as shown in (**c**). "Old" 60S ribosomal subunits are shown in red, "new" 60S ribosomal subunits are shown in green. Scale bar = 10 μm. **e** Quantification of Ribo-Halo experiments exemplified in (**d**). The nuclear vs cytoplasmic intensity of newly synthesized ribosomes (left) or the cytoplasmic intensity of new vs old ribosomes in the cytoplasm (right) was determined. Two-sided *t*-testing was performed for statistical analysis. Experiments were performed with four replicates and the data are shown as mean ± standard deviation. Source data for (**a**), (**b**), and (**e**) are provided as a Source Data file.

phase cells more strongly than their individual depletions (Supplementary Fig. 6c).

CyclinD-CDK4/6 primarily trigger G1/S transition by phosphorylating the retinoblastoma tumor suppressor (pRB) thereby controlling the activity of E2F family members. To investigate whether SENP3 influences the pRB-E2F axis, we compared the phosphorylation status

of pRB in p53-deficient RPE1 cells upon either individual and combined depletion of CDK4/6 or depletion of SENP3 alone or combined with CDK4 (Fig. 7c and Supplementary Fig. 6d). Individual depletion of either CDK4 or CDK6 reduced phosphorylation of pRB at S807/811 and T826, while co-depletion of both kinases completely abrogated phosphorylation. Importantly, depletion of SENP3 reduces pRB

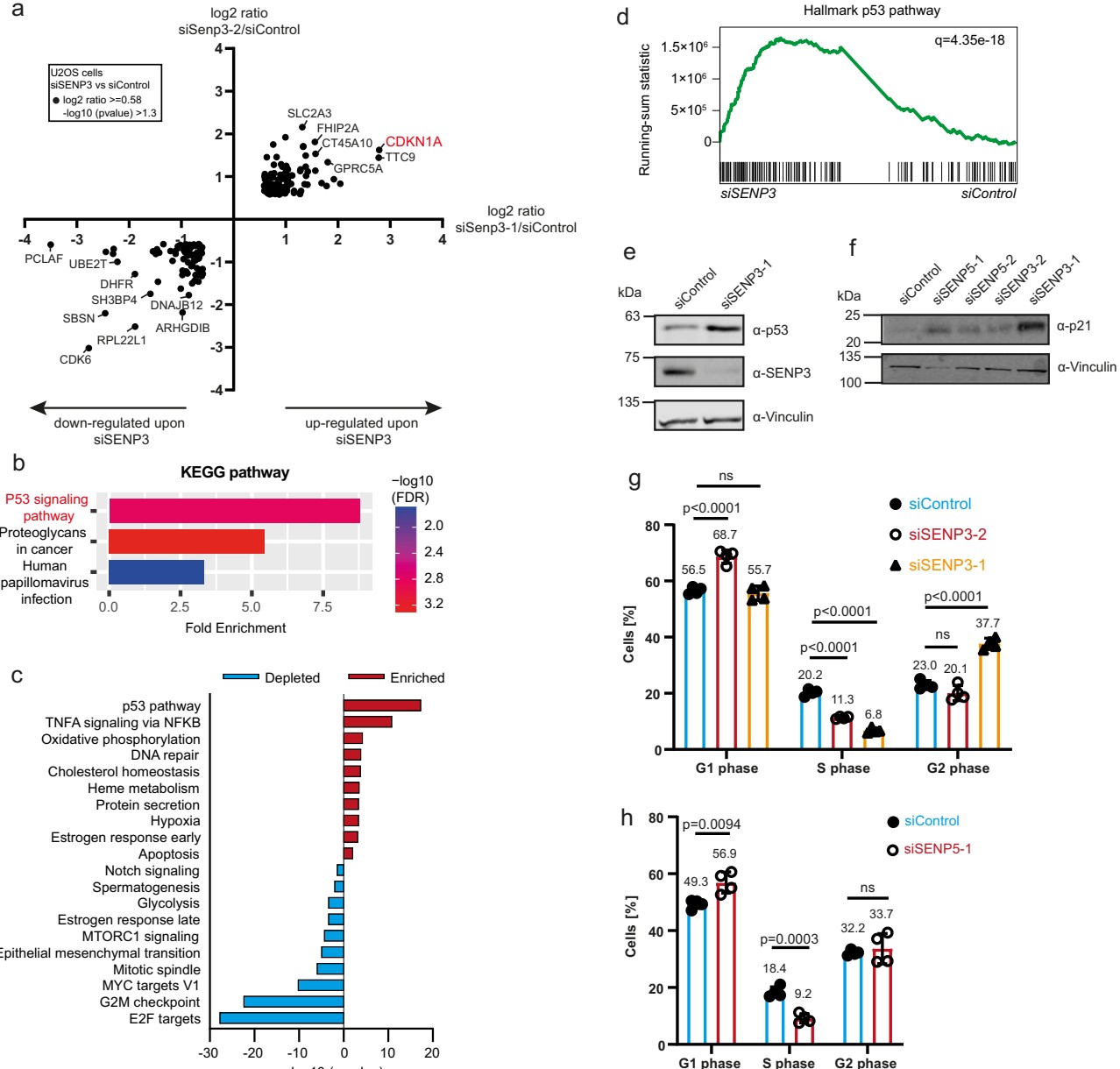

**Fig. 5 | SENP3 and SENP5 deficiency triggers the canonical p53-dependent impaired ribosome biogenesis checkpoint. a** Whole-cell proteome of U2OS cells transfected with siSENP3-1 (*X*-axis) or siSENP3-2 (*Y*-axis) compared to control siRNA transfection. Results of the TMT-based MS analysis are visualized in a XY diagram, comparing either of the two SENP3 siRNAs against the control. Only significant hits that are regulated in the same way with both siRNAs are displayed (log2 ratio ≥ 0.58; −log10 *p* value was ≥1.3). The identification of those candidates was based on two-sided Student's *t*-test analysis comparing the normalized TMT abundances of siSENP3 with siControl. Experiments were performed with four replicates. **b** KEGG pathway enrichment analysis of regulated proteins identified in (**a**) (log2 ratio ≥ 1, −log10 *p* value ≥ 1.3). Analysis was done using the ShinyGO tool, applying an FDR cutoff of 0.05 (**c**) top 10 analysis (**d**) Gene Set Enrichment Analysis (GSEA) of Hallmark analysis of RNA-Seq analysis performed in U2OS WT cell transfected for 72 h with two different siRNAs against SENP3 or control siRNA. Enriched terms upon SENP3 KD are shown in red, depleted terms are colored in blue. **e** U2OS cells

were transfected with siSENP3 or siControl 72 h prior to cell lysis. Lysates were immunoblotted as indicated. **f** U2OS cells were depleted for SENP3 or SENP5 with two independent siRNAs for 72 h and subsequently immunoblotted as indicated. **g** Cell cycle analysis of U2OS cells transfected with two different siRNAs against SENP3 (red and orange) or siControl (blue) for 72. PI staining was performed and the cell cycle profile was analyzed with flow cytometry. Subsequently, the percentage of cells in G1, S or G2 phase was determined. Significance level was calculated by two-sided *t*-tests. Experiments were performed with four replicates and are shown as mean ± standard deviation. Mean values of the percentages are shown above the bars. **h** PI staining of U2OS cells transfected with siSENP5 (red) or siControl (blue) for 72 h. Cell cycle profile was analyzed by flow cytometry and the percentages of cells in either G1, S or G2 phase were determined. Significance level was calculated by two-sided *t*-tests. Experiments were performed with four replicates and are shown as mean ± standard deviation. Mean values of the percentages are shown above the bars. Source data for (**e**)−(**h**) are provided as a Source Data file.

phosphorylation at these sites, thereby phenocopying CDK6 depletion. Combined CDK4/SENP3 depletion further dampens pRB phosphorylation recapitulating lack of CDK4/6 (Fig. 7c and Supplementary Fig. 6d). Importantly, the reduced pRB phosphorylation upon SENP3

loss could be rescued by CDK6 re-expression indicating that impaired cell cycle progression upon depletion of SENP3 is directly linked to CDK6 and alterations in the phosphorylation status of pRB (Fig. 7d and Supplementary Fig. 6e).

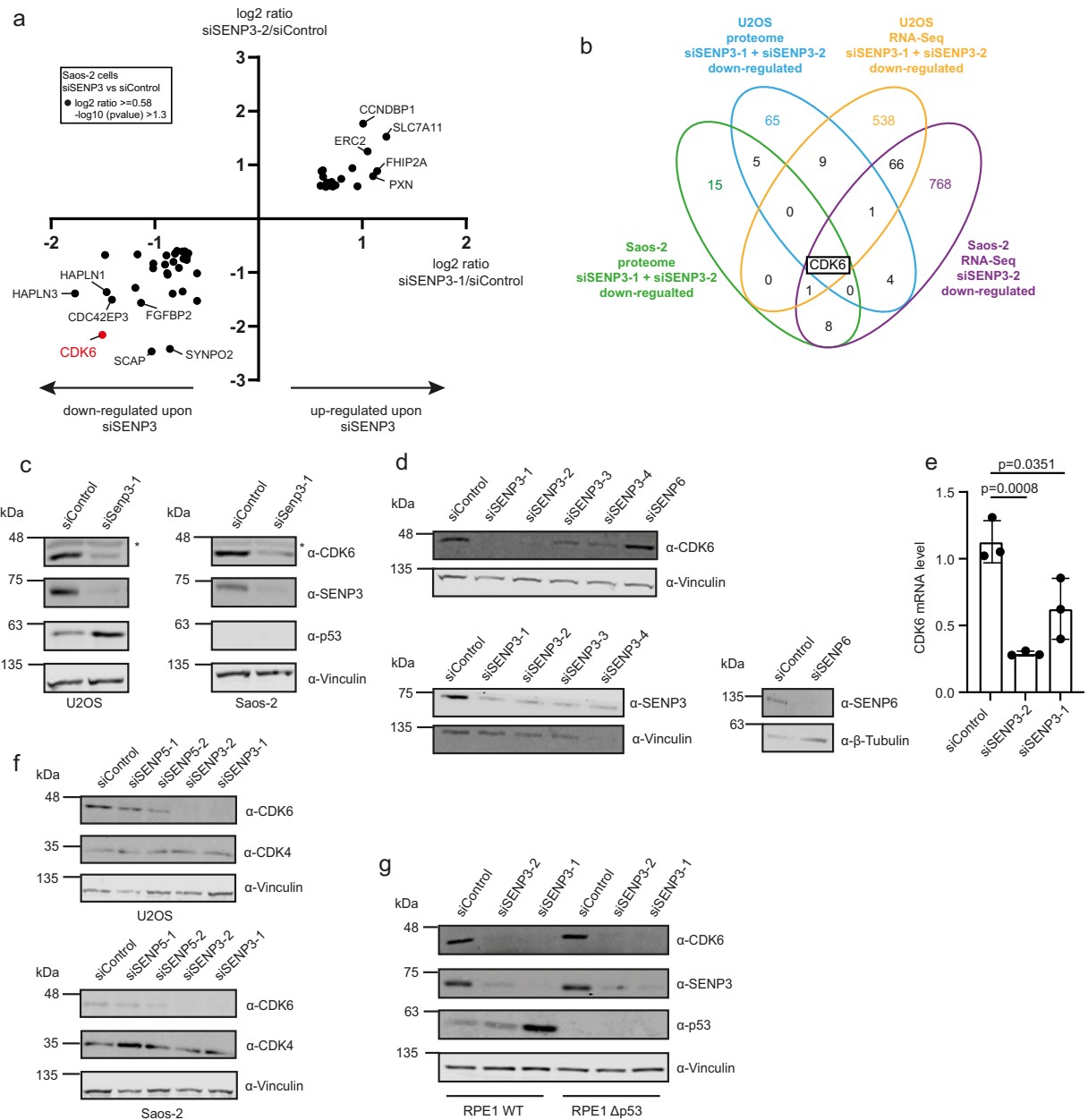

**Fig. 6 | SENP3 and SENP5 deficiency induces downregulation of CDK6. a** Whole-cell proteome of SAOS-2 cells transfected with siSENP3-1 (*X*-axis) or siSENP3-2 (*Y*-axis) compared to control siRNA transfection. Results of the TMT-based MS analysis are visualized in a XY diagram, comparing either of the two SENP3 siRNAs against the control. Hits were considered as significantly enriched, if the log2 ratio was ≥0.58 and the −log10 *p* value was ≥1.3. Only significant hits, that are regulated in the same way with both siRNAs are displayed. The identification of those candidates was based on two-sided Student's *t*-test analysis comparing the normalized TMT abundances of siSENP3 with siControl. CDK6 is highlighted in red. Experiments were performed with four replicates for siControl and siSENP3-2 and in triplicates for siSENP3-1. **b** Venn diagram showing the overlap of proteins, that were downregulated upon SENP3 knockdown in whole-cell proteome analysis of SAOS-2 and U2OS cells (Fig. 5) using two different SENP3 siRNAs (log2 ratio ≥ 0.58, −log10 *p* value ≥1.3). Furthermore, proteins identified as downregulated on mRNA level in transcriptomic analysis of U2OS cells or SAOS-2 cells, transfected with siSENP3-2, were considered. **c** U2OS (left panel) or SAOS-2 cell (right panel) were transfected with control siRNA or siRNA against SENP3. After 72 h, cells were lysed and proteins were analyzed by immunoblotting as indicated. **d** KD of SENP3 using four independent siRNAs, SENP6 KD or control KD were performed in U2OS cells. 72 h after transfection cell lysates were analyzed by immunoblotting as indicated. **e** SENP3 was depleted from U2OS cells by two different siRNAs. 72 h after transfection cells were lysed, total RNA was isolated and reverse transcribed into cDNA. By qPCR analysis the CDK6 mRNA level was determined. Three independent experiments were performed and are shown as mean ± standard deviation. Two-sided *t*-tests were performed. **f** U2OS (left panel) or SAOS-2 cells (right panel) were transfected with control siRNA or two independent siRNAs against SENP3 or SENP5. After 72 h, cells were lysed and proteins were analyzed by immunoblotting as indicated. **g** Parental RPE1 or RPE1$^{\Delta p53}$ cells were transfected with two different SENP3 siRNAs or siControl for 72 h. Subsequently, cells were lysed and the protein level of CDK6, SENP3, p53 and Vinculin was analyzed by immunoblotting. Source data for (**c**)–(**g**) are provided as a Source Data file.

Altogether, these data demonstrate that inactivation of SENP3 or SENP5 downregulates cellular CDK6 levels in both p53-proficient and p53-deficient cells. A reduced fraction of S phase cells together with impaired proliferation is fully consistent with reduced CDK6 expression.

## SENP3/5 represent a potential vulnerability in p53-mutant tumors

A subset of tumor cell lines depends on high CDK6 expression and targeting of CDK6 is an emerging anti-cancer strategy. To explore

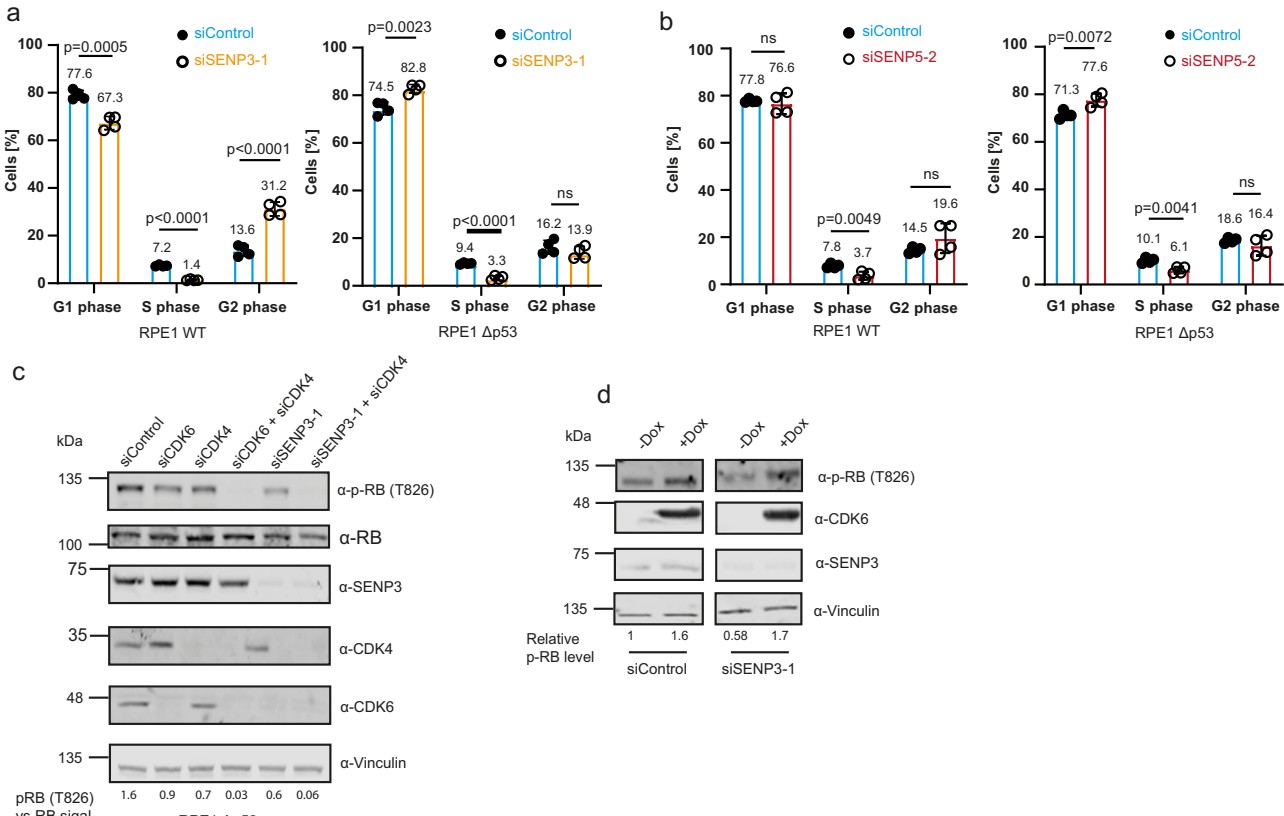

**Fig. 7 | SENP3 and SENP5 deficiency induces impairment of cell cycle progression in a CDK6/RB-dependent pathway. a** PI staining and cell cycle analysis by flow cytometry of parental RPE1 (left) or RPE1$^{\Delta p53}$ (right) cells transfected with siSENP3 (orange) or siControl (blue) for 72 h. Percentage of G1, S or G2 phase cells was determined. Significance was determined by two-sided t-tests. Experiments were performed with four replicates and are shown as mean ± standard deviation. Mean values of the percentages are added above the bars. **b** PI staining of SENP5 depleted parental RPE1 (left) or RPE1$^{\Delta p53}$ (right) cells performed as in (**a**). Significance level was calculated by two-sided t-testing. Experiments were performed with four replicates and are shown as mean ± standard deviation. Mean values of the percentages are added above the bars. **c** Immunoblot analysis of RPE1$^{\Delta p53}$ cells transfected for 72 h with siRNAs as indicated. Antibodies were used as indicated. The phospho-RB as well as the RB level were quantified using the LI-COR Image studio software. The ratio of pRB level against the whole RB level is indicated by the respective numbers. **d** U2OS cells expressing FLAG-CDK6 under a doxycycline-inducible promoter were transfected with control siRNA or siRNA against SENP3 for 72 h. FLAG-CDK6 expression was induced by supplementing the media with doxycycline (0.05 μg/ml) 48 h before cell lysis. Immunoblots were performed as indicated. phospho-RB (T826) levels were quantified and normalized to non-induced conditions. Source data for (**a**)–(**d**) are provided as a Source Data file.

whether SENP3/5 might be exploited to target CDK6, we used the pancreatic cancer cell line BxPC3 as a high CDK6 model system that concomitantly exhibits low CDK4 expression and additionally lacks functional p53 (Fig. 8a). Depletion of SENP3 from BxPC3 cells significantly reduces pRB phosphorylation again recapitulating CDK6 depletion (Fig. 8a). Depletion of SENP3 or SENP5 from BxPC3 cells impairs cell cycle progression as indicated by a reduced number of cells in S phase (Fig. 8b). Co-depletion of SENP3/5 further reduces S phase cells again indicating a redundancy of both isopeptidases. To expand on this data, we performed analogous experiments in Ramos cells, a CDK6-dependent, p53-mutant B cell model. Depletion of SENP3 or SENP5 downregulates CDK6 and affects cell proliferation as indicated by a reduced number of S phase cells (Fig. 8c, d). Altogether, these data indicate that SENP3/5 represent a potential vulnerability in p53-mutant tumors.

**Impaired ribosome biogenesis generally affects CDK6 expression**

Given the functional implications of SENP3/5 in ribosome biogenesis, we explored whether impaired ribosome biogenesis is generally linked to CDK6 signaling. To this end, we treated U2OS or SAOS-2 cells with the RNA polymerase I inhibitor CX-5461 and monitored CDK6 expression by immunoblotting. In U2OS cells, CDK6 was strongly downregulated, while p53 levels were induced.

Importantly, SAOS-2 cells also exhibit significantly reduced CDK6 levels upon exposure to CX-5461 (Fig. 9a), supporting the notion that impaired ribosome biogenesis induces the downregulation of CDK6 in a p53-independent process. To strengthen this idea, we used siRNAs to deplete well-characterized ribosome biogenesis factors from p53-proficient (U2OS and RPE1) or p53-deficient (SAOS-2 and RPE1$^{\Delta p53}$) cells (Fig. 9b, cstr, Supplementary Fig. 7a–c). The selected ribosome biogenesis factors are involved in nuclear or cytoplasmic steps of either 40S (TSR1, DHX37, SENP3, SENP5) or 60S (LAS1L, PES1, SENP3, SBDS) maturation. Importantly, when compared to at least three unrelated control siRNAs (siControl, siSENP6, siWDR5), depletion of any of the selected *trans*-acting ribosome assembly factors strongly reduced CDK6 levels in both the p53-proficient and the p53-deficient RPE1 cells, indicating that perturbation of ribosome biogenesis generally converges on the downregulation of CDK6 (Fig. 9b, c). In further support of these data depletion of a set of RPS or RPL proteins triggers the loss of CDK6 (Fig. 9d). Notable exceptions are RPS7 and RPL11, where depletion does not affect CDK6, pointing to a specific function of RPS7 and/or RPL11 in signaling impaired ribosome biogenesis to the CDK6-pRb axis (Supplementary Fig. 7d). Altogether, these data for the first time demonstrate that the downregulation of CDK6 is a general response to the perturbation of ribosome biogenesis in tumor cells, irrespective of their p53 status (Fig. 9e).

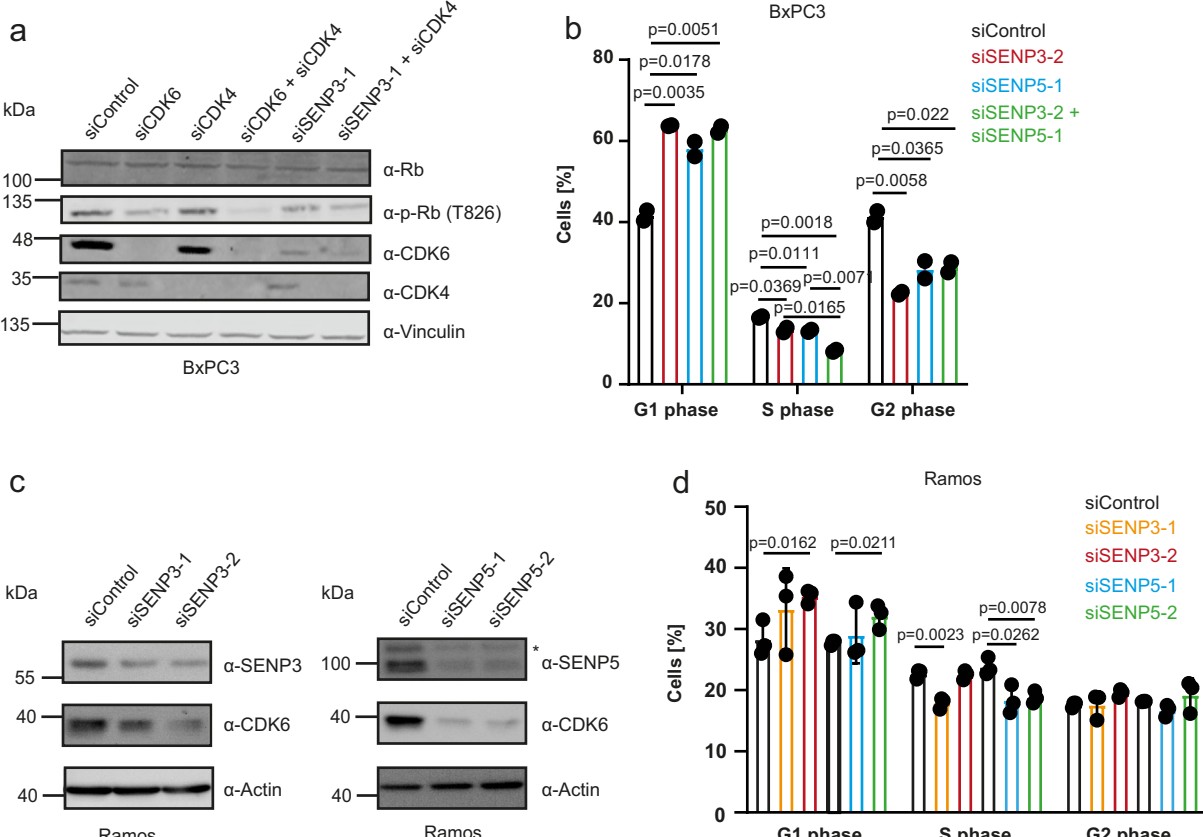

**Fig. 8 | SENP3 or SENP5 deficiency triggers CDK6 loss and cell cycle arrest in BxPC3 and Ramos cell lines. a** BxPC3 cells were transfected with siRNAs as indicated. 72 h after transfection cells were lysed and proteins were analyzed by immunoblotting as indicated. **b** Cell cycle analysis by PI staining of BxPC3 cells depleted for SENP3 (red), SENP5 (blue), SENP3/5 (green) or siControl (black) for 72 h. Percentage of cells in the G1, S or G2 phase was determined and significance level was calculated using two-sided *t*-testing. Experiments were performed with two replicates and are shown as mean ± standard deviation. **c** Depletion of SENP3 or

SENP5 with two different siRNAs in Ramos cells for 72 h. Cell lysates were analyzed by immunoblotting as indicated. **d** PI staining of Ramos cells transfected with two independent siRNAs against SENP3 (orange, red), SENP5 (blue, green) or siControl (black). Percentage of cells in G1, S or G2 phase was determined and significance level was determined by two-sided *t*-testing. Experiments were performed with three replicates and are shown as mean ± standard deviation. Source data for (**a**)–(**d**) are provided as a Source Data file.

## Discussion

Ribosome biogenesis is a complex and energy-demanding process requiring tight coordination with cell growth and proliferation. Impairment of ribosome biogenesis activates a well-defined cell cycle checkpoint that primarily relies on the activation of p53 signaling. However, there is mounting evidence that p53-independent signaling networks connect impaired ribosome biogenesis to cell-cycle checkpoints. So far, however, these pathways have remained largely enigmatic. By characterizing the nucleolar SUMO deconjugases SENP3 and SENP5, we found that both proteases control the SUMOylation state of specific ribosome biogenesis factors and regulate 60S (SENP3) and 40S (SENP3/5) ribosome maturation pathways. Accordingly, inactivation of SENP3 or SENP5 induces a canonical p53-mediated cell cycle arrest. However, we discovered that inactivation of SENP3 or SENP5 strongly and specifically downregulates the expression of the key cell cycle regulator CDK6 in a p53-independent process. Accordingly, depletion of SENP3 or SENP5 impairs G1/S transition and cell proliferation in both p53-proficient and p53-deficient cells. Importantly, we further revealed that impaired ribosome maturation induced by depletion of a panel of ribosomal proteins or ribosome biogenesis factors or by chemical inhibition of RNA polymerase I, generally triggers loss of CDK6, independent of the cellular p53 status. Altogether, our data unveil a long-sought p53-independent checkpoint of impaired ribosome biogenesis (see our model Fig. 9e).

The six members of the human SENP family exhibit a pairwise evolutionary relationship and sequence similarity, with SENP1-SENP2, SENP3-SENP5 and SENP6-SENP7 being most closely related[11]. SENP3 and SENP5 share almost 60% identity within their catalytic domain spanning the 250 C-terminal residues. Furthermore, both isopeptidases preferentially act on SUMO2/3 conjugates and are compartmentalized in the nucleolus. So far, however, it was unclear to what extent SENP3 and SENP5 exert redundant or specific functions and whether they act on distinct or common substrates. We and others have previously demonstrated that SENP3 is part of the mammalian PELP1 complex and revealed that this complex is the functional counterpart of the *Saccharomyces cerevisiae* Rix1 complex[14,28]. The human PELP1 complex—also termed rixosome—comprises the core components PELP1, WDR18, TEX10 and the associated factors MDN1, LAS1L and NOL9. The complex acts on early pre-60S particles, where the endonuclease LAS1L mediates pre-rRNA cleavage within the ITS2 region that separates the 5.8S rRNA from the 28S rRNA[29]. ITS2 processing by LAS1L is coordinated by PELP1 and the large AAA-ATPase MDN1 (Rea1 in yeast), which mediates crucial re-modeling steps on pre-60S particles. Based on data from yeast, it has been proposed that the Rix1-Rea1 re-modeling machinery triggers the correct timing of ITS2 processing[30]. In our previous work, we have shown that balanced SUMO conjugation-deconjugation coordinates the timely association of MDN1 with the PELP1 complex[15]. We proposed a model, where SUMOylated PELP1 recruits MDN1, while deconjugation by SENP3 is

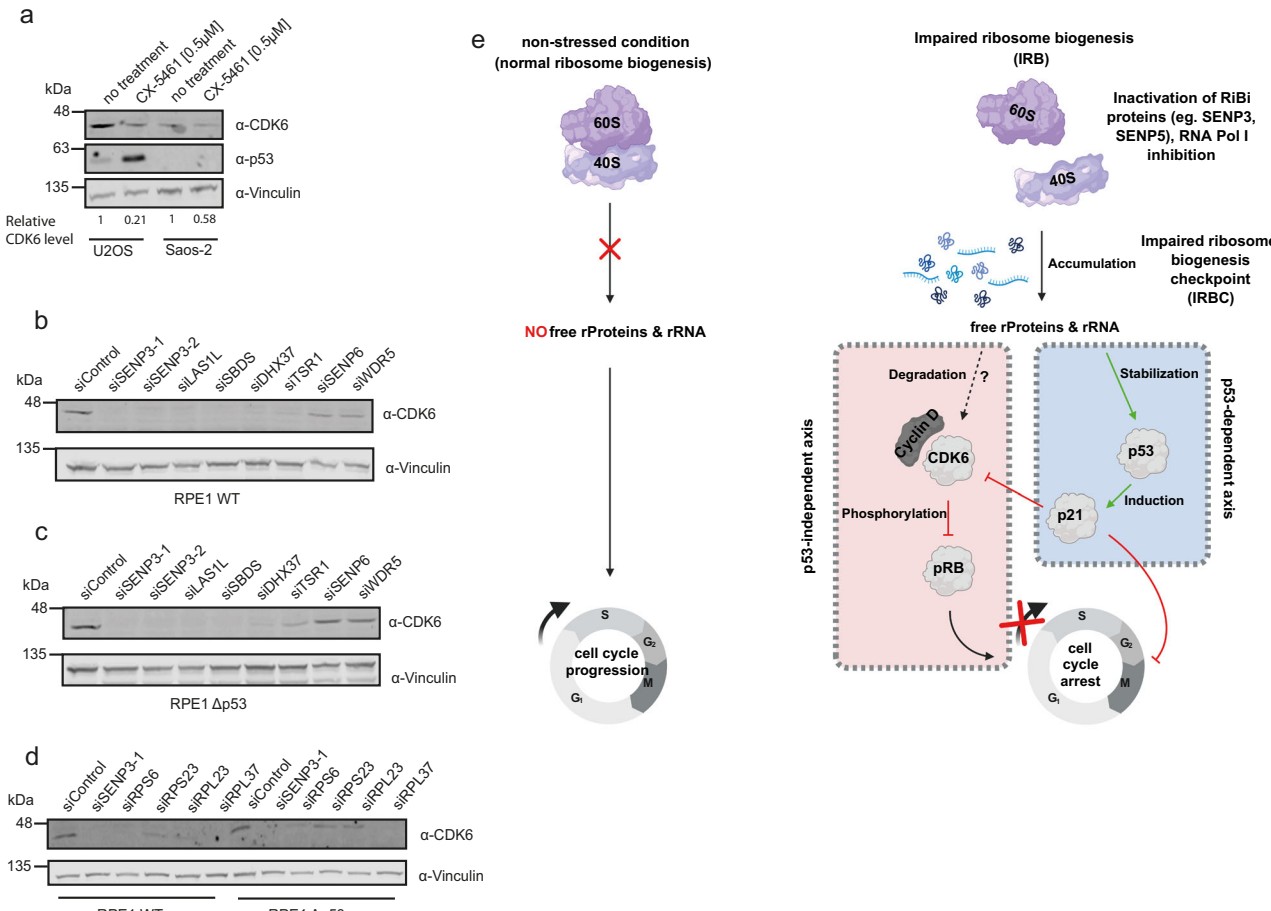

**Fig. 9 | Downregulation of CDK6 is a general response to disturbed ribosome biogenesis. a** U2OS and SAOS-2 cells were treated overnight with 0.5 μM CX-5461. After cell lysis immunoblotting was performed as indicated. CDK6 protein levels were quantified using the LI-COR Image studio software, normalizing the CDK6 signal to Vinculin. **b, c** siRNA transfection of parental RPE1 (**b**) or RPE1$^{\Delta p53}$ (**c**) cells with the indicated siRNAs. 72 h post transfection cells were lysed and proteins

were analyzed by immunoblotting against CDK6 and Vinculin. **d** Parental RPE1 or RPE1$^{\Delta p53}$ cells were transfected for 72 h with siRNAs as indicated. Cell lysates were immunoblotted against CDK6 or Vinculin. **e** Model depicting the activation of the canonical p53-dependent and the non-canonical CDK6-dependent IRBC. Activation of both pathways leads ultimately to cell cycle arrest. The model was created using BioRender. Source data for (**a**)–(**d**) are provided as a source data file.

needed for its release. Accordingly, inactivation of SENP3 traps MDN1 on pre-60S particles. Our unbiased proteomics approach now validates the role of SENP3 in countering PELP1 modification, but goes beyond previous findings. The new data provide evidence that SENP3 counters not only the SUMOylation of LAS1L and PELP1, but also limits modification of additional components, including MDN1, TEX10 and NOL9. This is in line with the general concept that the SUMO conjugation and deconjugation machinery acts on entire groups of proteins that are often found in protein complexes[31,32]. Based on the new data, we hypothesize that the unrestricted SUMOylation of multiple members of the PELP1 complex cooperates to lock the complex on pre-60S particles. We anticipate that this is due to the "glue-like" properties of multi-valent SUMO-SIM (SUMO-interacting motif) contacts between several subunits of the complex[33]. This interpretation is consistent with the enhanced pre-60S association of PELP1 observed here in cells depleted of SENP3/5. Interestingly, the rixosome also functions in Polycomb-mediated silencing and recent work demonstrated that SENP3 is required for Polycomb-rixosome interactions underlining the importance of SENP3 in controlling protein-protein interactions[34,35].

In addition to the identification of additional substrates at pre-60S particles, we show here that SENP3 also functions in 40S ribosomal subunit maturation. UTP14A was identified as an interaction partner and target of SENP3. Many early ribosome biogenesis factors form distinct sub-complexes, including the U3 snoRNP and the UTP-A/B/C

modules. The U3 snoRNP guides critical rRNA folding steps within the SSU, which ultimately allows pre-rRNA cleavage events that separate the pre-60S particle from the pre-40S. Formation of pre-40S particles requires dismantling of the 90S pre-ribosome triggered by the RNA helicase DHX37, whose activity is stimulated by its cofactor UTP14A[36,37]. Our data demonstrate that constitutive SUMOylation of UTP14A triggered by the absence of SENP3/5 affects its association with pre-40S particles, thereby preventing critical re-modeling steps[38,39]. Based on the available cryo-EM structures, we hypothesize that in case of UTP14A, the attachment of SUMO at the major SUMOylation site K733 sterically hinders the incorporation of UTP14A in pre-ribosomes[24,25]. Our functional data further indicate that SENP3 and SENP5 exert partially redundant functions in 40S maturation, whereas 60S maturation is primarily controlled by SENP3. Partial redundancy between SENP3 and SENP5 is also inferred from the observation that the constitutive knock-out of SENP3 by CRSIPR/Cas-mediated gene inactivation is accompanied by the upregulation of SENP5 at both the RNA and protein level, likely explaining the lack of an obvious growth defect upon constitutive inactivation of SENP3 (Supplementary Data 3).

By contrast, transient siRNA-mediated depletion of SENP3 in p53-proficient U2OS cells or RPE1 cells by several independent siRNAs induces a p53 response and impairs cell cycle progression. These data confirm previous findings and are consistent with the activation of the

canonical p53-dependent IRBC[28]. Similarly, loss of SENP5 induces a p53 response, albeit to a lesser extent. This again can be explained by the partial redundancy of both proteins and the generally lower levels of SENP5 in U2OS or RPE1 cells when compared to SENP3.

By exploring changes in cellular signaling pathways upon depletion of SENP3 in cells with deleted or non-functional p53, we observed a reduction of the cell-cycle regulator CDK6 at both the mRNA and protein levels. As CDK6—together with CDK4—is the critical regulator of G1/S transition, this explains the reduced fraction of S phase cells upon depletion of SENP3. Depletion of SENP5 generally phenocopies these effects, albeit with less drastic outcomes, which is consistent with the concept of SENP3 taking over SENP5 functions. Most importantly, subsequent experiments revealed that perturbation of ribosome biogenesis generally impacts CDK6 expression in both p53-proficient and deficient cells. RNAi-mediated depletion of established trans-acting ribosome assembly factors that either affect the pre-40S (TSR1, DHX37, SENP3 or SENP5) or the pre-60S (LAS1L, PES1, SENP3, SBDS) pathway converges on CDK6 downregulation. Furthermore, inhibition of RNA polymerase I by the chemical inhibitor CX-5461 or depletion of RPS/RPL proteins reduces CDK6 levels. From these data, we conclude that loss of CDK6 is a common response triggered upon perturbation of ribosome biogenesis. We propose that down-regulation of CDK6 is a central node in the long-sought p53-independent checkpoint of impaired ribosome biogenesis. p53-independent signaling pathways that connect alteration of ribosome biogenesis with cell cycle progression have already been described in lower eukaryotes that lack p53[40]. For example, balanced production of ribosome components was shown to be required for proper G1/S transition in *Saccharomyces cerevisiae*[41]. It was further shown that perturbation of ribosome biogenesis inhibits G1/S passage in a Whi5-dependent mechanism[42]. Whi5 is the yeast functional equivalent of the retinoblastoma (pRB) protein, the mammalian key target of CDK4/6. In the canonical pRB-E2F signaling axis, phosphorylation of pRB by CDK4/6 relieves the inhibitory constraint of pRB on E2F thereby promoting G1/S transition. Reduced CDK6 will therefore potentiate the inhibitory function of pRB on E2F. These data indicate that p53-independent IRBC signaling is an evolutionary conserved pathway that impinges on pRB/Whi5. In support of this idea, it has been shown that ribosome biogenesis defects contribute to cell cycle arrest through the pRB pathway in senescent cells[43]. Mechanistically, it has been shown that RPS14 accumulates in the soluble non-ribosomal fraction of senescent cells, where it directly binds to and inhibits CDK4. This exemplifies how unincorporated ribosomal proteins or trans-acting factors, which are generated upon perturbed ribosome biogenesis, can function as critical signaling molecules in cell cycle control. In agreement with this idea, free, cytosolic UTP18 was shown to control the stability of the p21 mRNA[44]. Our data indicate that perturbation of ribosome biogenesis affects CDK6 expression at the transcript level, e.g., by either limiting its synthesis or altering its mRNA stability. The detailed signaling processes that connect impaired ribosome biogenesis to CDK6 alterations are currently unknown, but we anticipate that unassembled ribosomal proteins or trans-acting factors impinge on *CDK6* regulation. In line with this idea, we show that an imbalance of ribosomal proteins generated by the depletion of a set of ribosomal proteins downregulates CDK6. Notably, however, depletion of RPL11 or RPS7 does not affect CDK6 possibly indicating their involvement in signaling impaired ribosome biogenesis to the CDK6-pRB axis. Interestingly, binding of RPL11 to MDM2 not only releases it from p53, but also from E2F, where it acts as a transcriptional co-activator on E2F target genes potentially including CDK6. Furthermore, RPL11 also recruits *miR-24* and the microRNA-induced silencing complex (miR-ISC) to the *MYC* RNA transcript regulating its turnover to repress *MYC* expression in response to ribosomal stress[45]. Whether and how RPL11 or RPS7 are involved in p53-independent IRBC signaling via CDK6 remains to be determined.

Irrespective of the detailed underlying molecular mechanisms our findings pave the way for exploiting this pathway in cancer cells, which are dependent on CDK6–cyclin D3 complexes[46]. We indeed show that in two CDK6-dependent tumor cell models, depletion of SENP3 or SENP5 downregulates CDK6 and affects cell proliferation. Exploiting this so far unrecognized checkpoint by targeting ribosome biogenesis factors, such as SENP3 and SENP5, could thus represent a powerful strategy to inactivate CDK6 in human tumors irrespective of their p53 status.

## Methods

### Cell culture and transfection
U2OS (female, ATCC HTB-96), SAOS-2 (female, ATCC HTB-85), HeLa (female, ATCC CCL-2), HEK293T (female, ATCC CRL-3216) RPE1 (female, ATCC CRL-4000), BxPC3 (female, ATCC CRL-1687) and Ramos (male, ATCC CRL-1596) cells were purchased from ATCC and cultured under standard conditions. Generation of endogenously tagged HeLa cells (RPS3-Halo and RPL28-Halo) and CRISPR KO cell lines (RPE1$^{\Delta p53}$, U2OS$^{\Delta SENP3}$) was done as described below. The pRTS1 episomal expression plasmid was used to generate conditional CDK6-expressing U2OS cells (Bornkamm et al.)[12,47]. Cells were transfected with Fugene and selected by adding 400 μg/ml hygromycin for 11 days to the growth media. CDK6 expression was induced by addition of 0.05 μg/ml doxycycline. The calcium phosphate method was used for transient plasmid transfection of HEK293T cells. Cells were harvested 48 h after transfection. Plasmids used in this study are listed in Supplementary Data 12. For adherent cells, siRNA transfection was done by using Lipofectamine RNAiMAX reagent (Thermo Fisher Scientific) for 72 h, if not stated differently. siRNAs used in this study are listed in Supplementary Data 13. Ramos cells were transfected sequentially by electroporation at 0 h and after 24 h using the Neon™ Transfection System (Invitrogen) according to the manufacturer's manual and the settings 1350 V; 30 ms; 1Pulse. Transfected cells were analyzed 72 h after the first electroporation. MG-132 (25 μM) was given to the cells 4 h prior to lysis. The SUMO inhibitor, TAK-981 was used for 4 h at a concentration of 100 nM. CX-5461 was used overnight at a concentration of 0.5 μM.

### Proliferation assay
Parental RPE1 or RPE1$^{\Delta p53}$ cells were reversely transfected with siRNA and immediately 2000 cells were seeded into an E-Plate 96 (OMNI Life Sciences). Plates were incubated at 37 °C for 100 h while the proliferation was continuously monitored by the XCELLigence RTCA SP instrument (OMNI Life Science). During this time, the impedance is measured by the device and displayed as "Cell Index". Background impedance was determined for media without cells. Each experiment was performed as duplicate measurements and the mean value was determined.

### CRISPR methods
Endogenously tagged HeLa RPS3- and RPL28-Halo (Ribo-Halo cells) were generated using the CRISPR-Cas12a-assisted PCR-tagging system according to Fueller et al.[48]. In brief, cells are transfected with a Cas12-encoding plasmid (pcDNA3.1-hAsCpf1 (TYCV)) together with a PCR product containing the gRNA, the repair template for the designated genomic locus and a selection marker. PCR primers for *RPS3* and *RPL28* were designed by using the Online Oligo Design tool (http://www.pcr-tagging.com) and PCR was performed on pMaCTag-Z23. Transfected HeLa cells were selected with 400 μg/ml Zeocin and single clones were raised. For the validation, cells were labeled overnight with 10 nM R110 Halo-ligand for 17 h and clones showing a positive signal were subjected to western blotting using endogenous antibodies for RPL28 and RPS3.

Following guide RNAs for ΔSENP3 cell lines were designed using the sgRNA selection tool of the Broad Institute (2016 edition): Guide 1

forward gggctccttactctgtacgc, guide 1 reverse gcgtacagagtaaggagccc, guide 2 forward cctccacctgacttgagtcg, guide 2 reverse cgactcaagt-caggtggagg, guide 3 forward cagcaatgtgtgcagcatcg, guide 3 reverse cgatgctgcacacattgctg. Virus was produced in HEK293T cells by pooling all sgRNAs against *SENP3*. Afterward, U2OS cells were transduced and gene inactivation was verified by anti-SENP3 immunoblotting (Supplementary Fig. 1c). All experiments were done with pools of infected cells.

## Immunoprecipitation
HEK293T cells were transfected with the calcium phosphate method. After 48 h, cells were collected on ice and lysed by rotation for 20 min at 4 °C (50 mM Tris/HCl, 150 mM NaCl, 1% NP40 [v/v], 1 mM EDTA, 0.1% sodium deoxycholate [w/v], pH 7.5). The buffer was supplemented freshly with protease inhibitors (1 mM PMSF, 2 μg/ml Aprotinin, 2 μg/ml Leupeptin, 1 μg/ml Pepstatin A). Lysates were cleared by centrifugation (20,000 g, 30 min, 4 °C) and incubated overnight with anti-FLAG beads (Sigma). After washing, immunoprecipitated proteins were eluted by boiling beads with SDS-PAGE loading buffer and separated by SDS-PAGE. Ni-NTA pull-down assays were done as described[49].

## Western blot analysis
Samples were separated by SDS-PAGE and transferred to NC membranes applying the wetblot technique using a Towbin buffer containing 20% methanol. Membranes were blocked in 5% milk dissolved in PBS-T before the primary antibody was added overnight (4 °C). After washing, secondary antibodies purchased from LI-COR (IRDye® 800CW Goat anti-Mouse or Rabbit IgG) were given to the membranes (1 h, RT) and the respective fluorescence signal was determined by the LI-COR system (Odyssey CLx Imager). All antibodies used in this study are listed in Supplementary Data 14.

## Sucrose density gradient centrifugation
HEK293 cells or HeLa cell lines were treated with 100 μg/ml cycloheximide for 10 min before to harvesting. Cells were resuspended in 20 mM HEPES pH 7.6, 100 mM KCl, 5 mM MgCl$_2$, 0.5% NP40, 100 μg/ml cycloheximide, 2 mM DTT, 0.625% Triton X-100, 0.625% deoxycholate supplemented with protease and RNase inhibitors and lysed on ice for 5 min. Cell debris was pelleted by centrifugation at 10,000 g for 10 min at 4 °C. Cleared cell extracts were loaded onto 10–50% sucrose gradients prepared in lysis buffer lacking detergents. After centrifugation in an SW-40Ti rotor at 35,000 rpm for 2.5 h, gradients were fractionated and an absorbance profile at 260 nm was generated using a BioComp Gradient Master. Proteins in each fraction were precipitated using 20% trichloroacetic acid before separation by SDS-PAGE and analysis by western blotting.

## Northern blot analysis
Total RNA was extracted from siRNA-treated cells using TRI Reagent (Sigma-Aldrich) according to the manufacturer's instructions. 5 μg of total RNA was separated by denaturing (glyoxal) agarose gel (1.2%) electrophoresis. RNAs were hydrolyzed in situ by treatment with 0.1 M NaOH and transferred to a nylon membrane in 6× SCC (150 mM NaCl, 15 mM sodium citrate) by vacuum blotting. RNAs were crosslinked to membranes, which were pre-hybridized in SES1 buffer (0.25 M sodium phosphate pH 7.0, 7% SDS, 1 mM EDTA) at 37 °C for 1 h. 5′ [$^{32}$P]-labeled DNA oligonucleotides complementary to specific pre-rRNA sequences (5′ ITS1 5′-CCTCGCCCTCCGGGCTCCGTTAATGATC-3′, ITS2 5′-GCTCTC TCTTTCCCTCTCCGTCTTCC-3′, actin mRNA 5′-AGGGATAGCACAGC CTGGATAGCAAC-3′) were added and incubated for >14 h at 37 °C. Probes were removed and membranes were washed for 30 min each at 37 °C in 6× SSC and then 2× SSC supplemented with 0.1% SDS. Membranes were exposed to phosphorimager screens and signals detected using a Typhoon FLA9500. Pre-rRNA levels were quantified using ImageQuant software.

## PI staining and cell cycle analysis by flow cytometry
Parental RPE1, RPE1$^{Δp53}$, BxPC3, Ramos or U2OS cells were grown in 6-well plates under standard conditions and transfected with siRNA for 72 h. Cells were trypsinized and washed with PBS. Afterward, cells were fixed using an ice-cold 70% [v/v] EtOH solution and incubated for 1 h on ice. Following an additional washing step with PBS, cells were incubated with FACS-stain solution (0.5% Triton X-100, 20 μg/ml PI, 20 μg/ml RNase A in PBS) for 1 h to stain the DNA and digest remaining RNA. DNA content was measured by flow cytometry (BD FACSCanto™ II, BD Biosciences). Data were analyzed by FlowJo (FlowJo, LLC).

## qPCR and RNA-Seq
Total RNA was isolated using the High Pure RNA isolation kit (Roche). 500 ng total RNA was taken for cDNA synthesis using the Transcriptor First Strand cDNA Synthesis Kit (Roche). RT-qPCR was performed with qPCR SYBRGreen Master Mix (Steinbrenner) and KiCqStart™ Primers (Merck) using the LightCycler 480 II (Roche). *GAPDH* gene expression was used for normalization. The Delta-Delta Ct ($2^{-ΔΔCt}$) method was applied for the determination of respective mRNA expression. Primer efficiencies were determined as described[50]. Three independent experiments were performed in duplicates, using four technical replicates each.

RNA sequencing analysis has been described previously[51]. Total RNA was isolated as stated before with $n = 4$ biological replicates of each condition. Quality and amount were checked using TapeStation RNA (Agilent) and Qubit (Invitrogen). Processing, library generation and paired-end sequencing (150 bp/read) of RNA samples has been performed by Novogene (Cambridge, UK). Sequencing of samples has been performed using a HiSeq2500 Illumina device (RRID:SCR_020123) with a read depth of more than 20 M reads. Subsequent quality control, performed using FastQC (RRID:SCR_014583), data were analyzed using the Galaxy platform, using the local server version of usegalaxy.org (RRID:SCR_006281). Adapter sequences were removed from FASTQ files using Trimmomatic (RRID:SCR_011848)[52]. Final transcript quantification is described in the section Bioinformatic tools. The generated results were visualized with GraphPad Prism 8. RNA-Seq data have been stored at the European Nucleotide Archive and are accessible via accession ID: PRJEB57219.

## Immunofluorescence microscopy and microscopy of fluorescently labeled Ribo-Halo cells
Localization of FLAG-tagged catalytically dead SENP3 or SENP5 was analyzed in HEK293T cells after 48 h of plasmid transfection. Changes in UTP14A localization upon SENP3 or SENP5 knockdown were monitored in Hela cells after 72 h of siRNA treatment. To display the nucleoli, cells were co-stained with anti-BMS1. Cells were fixed in 4% PFA for 15 min followed by permeabilization for 10 min using 0.5% Triton X-100 in PBS. After blocking with BSA for 20 min, the respective primary antibody was added and incubated for 1 h. Subsequently, the slides were washed three times with PBS and the fluorescently labeled secondary antibody together with 1 μg/ml DAPI were added for 1 h. Finally, the coverslips were mounted on microscope slides using Pro-Long Gold Antifade reagent (Invitrogen).

Endogenously tagged HeLa RPS3- and RPL28-Halo were seeded on coverslips and knockdown was performed as described above. Labeling with HaloTag fluorescent ligands R110 and TMR (Promega) was performed according to An et al.[26] with some modifications to improve the signal differences between KDs. In brief, labeling of "old" ribosomes was done by incubating cells for 1 h with direct ligand 1 (100 nM) 24 h after performing the KD. After extensive washing, cells were incubated with the non-fluorescent HaloTag blocker 1-chloro-6-(2-propoxyethoxy)hexane (CPXH, CID 63684368 AKos Consulting & Solutions, GmbH) for 24 h (1 μM)[53]. Twenty-four hours before fixation, the non-fluorescent Halo-blocker was washed out and "new" ribosomes are labeled with ligand 2 (50 nM) until fixation of the cells.

Before fixation with 4% PFA, cells were washed with PBS. Subsequently, cells were incubated with 1 µg/ml DAPI and mounted on microscope slides using ProLong Gold Antifade reagent (Invitrogen).

Images were taken with a Leica TCS SP8 confocal microscope and processed using the Fiji-BioVoxxel bundle in ImageJ.

## Mass spectrometry

The identification of SENP3 and SENP5 interactors was performed by transfecting HEK293T cells with FLAG-tagged SENP3[C532S], SENP5[C713S] or respective MOCK control as described before[54]. The subsequent IP was done in triplicates.

SENP3 or SENP5 targets were enriched as described by Barysch et at.[55]. For SENP3 targets each IP (anti-SUMO2/3 or anti-IgG control) was done in triplicates using 8 mg of protein, while the identification of SENP5 targets was done with four replicates using 17 mg of protein. Afterward, IP samples were separated by SDS-PAGE, digested and purified as stated before[54]. Final proteomic analysis was performed as described[55]. Proteomic analyses were performed on an Easy nLC 1200 system (Thermo Fisher) coupled to Q Exactive HF mass spectrometer (Thermo Fisher). The mass spectrometer was operated in a data-dependent mode (MS1 scan range, 300–1750 m/z). Full-scan MS spectra of IPs/proteomic samples were acquired using 3E6 as an AGC target with a resolution of 60,000 at 200 m/z with a maximum injection time of 20 ms. The 15 most intense ions were fragmented by high collision-induced dissociation (HCD). Resolution for MS/MS spectra was set to 30,000/15,000 at 200 m/z, AGC target to 1E5, maximal injection time to 64 ms/25 ms. Data were deposited on PRIDE (Project Name: Identification of SENP3 and SENP5 target proteins by endogenous SUMO2/3 IP-MS; accession: PXD037793, PXD037796).

UTP14A interactomes were carried out in Hela cells transfected with indicated siRNAs for 72 h. For endogenous IPs cells were harvested on ice (50 mM HEPES, 150 mM NaCl, 1.5 mM MgCl₂, 1 mM EGTA, 10% Glycerol [v/v], 1% Triton X-100 [v/v], pH 7.2, 1 mM PMSF, 2 µg/ml Aprotinin, 2 µg/ml Leupeptin, 1 µg/ml Pepstatin A, 10 mM NEM) and lysed by rotation at 4 °C for 10 min. After removing of cell debris (20,000 g, 15 min, 4 °C), the lysates were pre-cleared by incubation with protein A/G PLUS-Agarose beads (Santa Cruz) for 1 h at 4 °C. IgG control or UTP14A antibody were crosslinked to the beads using Dimethyl pimelimidate (DMP) as described in ref. 55 before performing IPs overnight. Each IP was done with four replicates, using 4.5 mg of protein per IP. The following day, proteins were eluted by adding sodiumdeoxycholate (SDC) buffer (3% SDC, 50 mM Tris, pH 8.5) and incubating the samples at 95 °C for 5 min. Afterward, supernatants were reduced and alkylated by adding 1 mM TCEP and 4 mM chloroacetamide in 50 mM Tris pH 8.5. For subsequent protein digestion, using 500 ng Trypsin and 500 ng LysC, samples were diluted using 50 mM Tris pH 8.5 to reach a final SDC concentration of 1%. The digestion was stopped the following day by adding 0.25% TFA and peptides were subjected to styrene-divinyl benzene reverse phase sulfonate (SDB-RPS) polymer sorbent solid phase extraction STAGE tips (Kulak et al.)[56]. Dried peptides were resuspended in 2% ACN and 0.1% TFA and subjected to LC-MS analysis. The final proteomic analysis was performed on an easy nLC 1200 (Thermo Fisher). Peptides were eluted by non-linear gradient over 75 min and were afterward directly sprayed into a Fusion Lumos MS with a nanoFlex ion source (Thermo Fisher). Full-scan MS spectra (300–1500 m/z) of IP samples were acquired at a resolution of 60,000 at m/z 200, with an AGC target value of $4 \times 10^5$ and a maximum injection time of 50 ms. To obtain MS2 scans an isolation window of 1.4 Th and a maximum injection time of 54 ms were applied. Ions were fragmented by high energy collisional induced dissociation (HCD) using a collision energy of 30%. The resolution was set to 30,000 at m/z 200 and an AGC target value of $1.5 \times 10^5$ was applied. Data were deposited on PRIDE (Project Name: Identification of UTP14A interactors by endogenous UTP14A-IP-MS; accession: PXD043556).

Whole-cell proteome (WCP) analysis was accomplished in either Hela cells transfected with siSENP5 or siControl, SAOS-2 or U2OS cell lines transfected with siRNA against SENP3 (siSENP3-1 or siSENP3-2) or control siRNA. For each condition, four replicates were used. Due to the loss of one replicate during sample preparation (SAOS-2 siSenp3-1, replicate 3) only three replicates of this condition were considered for analysis. Seventy-two hours after transfection cells were rinsed three times with PBS and scraped in lysis buffer (2% SDS, 50 mM Tris/HCl, 10 mM TCEP, 40 mM CAA, 1 mM PMSF, 2 µg/ml Aprotinin, 2 µg/ml Leupeptin, 1 µg/ml Pepstatin A, pH 8.5). From this step on samples were kept in low-binding tubes (Eppendorf). To allow complete lysis samples were boiled and sonicated. Methanol and Chloroform were used to precipitate the proteins. In brief, four volumes ice-cold methanol and one volume ice-cold chloroform were added to the lysates and vigorously mixed. Next, three volumes of water were supplemented and samples were subjected to centrifugation (15,000 g, 4 °C, 15 min). The top layer was removed, three volumes ice-cold methanol were added and samples were vortexed. After centrifugation (15,000 g, 4 °C, 10 min) the protein pellet was washed two times with ice-cold methanol as mentioned before, transferred in a new tube and the pellet was air-dried. Thereafter the pellets were dissolved (8 M urea, 50 mM Tris/HCl, pH 8.2) at 37 °C for 30 min to allow determination of the protein concentration using BCA assay (Thermo). Samples were diluted to reach a urea concentration below 1.5 M. 50 µg of protein was digested overnight using Trypsin and LysC and stopped by addition of trifluoroacetic acid the next day. Afterward, samples were desalted using tC18 Sep-Pak SPE cartridges (Waters), dried by speed-vac and resolved in 200 mM EPPS, 10% ACN, pH 8.2. The microBCA assay kit (Thermo) was used to determine the peptide concentration prior to TMT labeling. 10 µg peptide per sample was supplemented with ACN to reach a final concentration of 20% and 1 µl (25 µg) of the respective TMTpro™ reagent (Thermo) was added. Successful labeling and equal mixing was tested by MS before samples were finally pooled, concentrated and desalted on STAGE tips as described[57]. The Pierce High pH Reversed-Phase Peptide Fractionation Kit (Thermo) was used to generate eight fractions. Proteomic analysis was performed on an easy nLC 1200 (Thermo Fisher). Peptides were eluted by non-linear gradient for each fraction over 165 min and afterward directly sprayed into a Fusion Lumos MS with a nanoFlex ion source (Thermo Fisher). Top-Speed method (1.5 s cycle time) with the RF lens at 30% was used for MS analysis. A resolution of 120,000 at m/z 200, a maximum injection time of 100 ms and an AGC target value of $4 \times 10^5$ were used to get full-scan MS spectra (350–1400 m/z). The Ion trap (Turbo) was used to obtain MS2 scans applying an isolation window of 0.7 Th and a maximum injection time of 50 ms. CID with a collision energy of 35% was used to achieve ion fragmentation. The 10 most intense MS2 fragment ions were used for SPS-MS3 analysis (isolation window 0.7 Th (MS1) and 2 Th (MS2)). Fragmentation of the ions was done using HCD with a normalized collision energy of 50%. For final analysis the Orbitrap was set to a scan range of 110–500 m/z, a AGC target value of $1.5 \times 10^5$, a resolution setting of 50,000 at m/z 200 and a maximum injection time of 86 ms. The dynamic exclusion time was set to 45 s and 7 ppm.

WCP analysis of HeLa cells transfected with siControl, siSENP3, siSENP5 or siSENP3/5, as control for UTP14A-IP-MS experiments, was done using three technical replicates. Sample preparation was carried out as described above. Peptides were fractionated using high-pH liquid-chromatography on a micro-flow HPLC (Dionex U3000 RSLC, Thermo Scientific). 45 µg of pooled and purified TMT labeled peptides resuspended in Solvent A (5 mM ammonium-bicarbonate, 5% ACN) were separated on a C18 column (XSelect CSH, 1 mm × 150 mm, 3.5 µm particle size; Waters) using a multistep gradient from 3 to 60% Solvent B (5 mM ammonium-bicarbonate, 80% ACN) over 65 min at a flow rate of 30 µl/min. Eluting peptides were collected every 43 s from minute 2 for 69 min into a total of 96 fractions, which were cross-concatenated into 24 fractions. Pooled fractions were dried in a vacuum

concentrator and resuspended in 2% ACN, 0.1% TFA for LC-MS analysis. Tryptic peptides were analyzed on an Orbitrap Ascend coupled to a VanquishNeo (Thermo Fisher Scientific) using a 35 cm long, 75 μm ID fused-silica column packed in house with 1.9 μm C18 particles (Reprosil pur, Dr Maisch), and kept at 50 °C using an integrated column oven (Sonation). HPLC solvents consisted of 0.1% Formic acid in water (Buffer A) and 0.1% Formic acid, 80% acetonitrile in water (Buffer B). Assuming equal amounts in each fraction, 400 ng of peptides were eluted by a non-linear gradient from 7 to 40% B over 90 min followed by a step-wise increase to 90% B in 6 min which was held for another 9 min. A synchronous precursor selection (SPS) multi-notch MS3 method was used in order to minimize ratio compression as previously described[58]. Full scan MS spectra (350–1400 m/z) were acquired with a resolution of 120,000 at m/z 200, maximum injection time of 100 ms and AGC target value of $4 \times 10^5$. The most intense precursors with a charge state between 2 and 6 per full scan were selected for fragmentation ("Top Speed" with a cycle time of 1.5 s) and isolated with a quadrupole isolation window of 0.7 Th. MS2 scans were performed in the Ion trap (Turbo) using a maximum injection time of 35 ms, AGC target value of 10,000 and fragmented using CID with a normalized collision energy (NCE) of 35%. SPS-MS3 scans for quantification were triggered only after a successful real-time search against the human canonical reference proteome from SwissProt with the same search parameter as stated below for data processing in Proteome Discoverer[59,60]. Criteria for passing the search were Xcorr: 2, dCn: 0.05 and precursor mass accuracy: 10 ppm (w/o second 10) . Maximum search time was 40 ms and peptide close-out was set to three peptides per protein. MS3 acquisition was performed on the 10 most intense MS2 fragment ions with an isolation window of 0.7 Th (MS) and 2 Th (MS2). Ions were fragmented using HCD with an NCE of 55% and analyzed in the Orbitrap with a resolution of 30,000 at m/z 200 (TurboTMT), scan range of 110–150 m/z, AGC target value of 150,000 and a maximum injection time of 59 ms. Repeated sequencing of already acquired precursors was limited by setting a dynamic exclusion of 60 s and 7 ppm and advanced peak determination was deactivated. All spectra were acquired in centroid mode.

Raw data analysis was achieved by using the MaxQuant software[61], with settings as described[54]. Analysis of the whole-cell proteomes was done using the Proteome Discoverer software (version 2.4) selecting SequenceHT node for database searches. The human trypsin digested proteome (Homo sapiens SwissProt database [20531]) was used for protein identification, while contaminants were spotted using MaxQuant "contaminants.fasta". TMTpro (K, +304.207 Da) at the N terminus and carbamidomethyl (+57.021 Da) at cysteine residues were set as fixed modifications, while TMTpro (K, +304.207 Da), methionine oxidation (M, +15.995 Da) and acetyl (+42.011 Da) at the N terminus were set as dynamic modifications. Fragment and precursor mass tolerance were set to 0.5 Da and 7 ppm, respectively. Quantification of reporter ions was done using default settings in consensus workflow. Details of the statistical analysis are described in quantification and statistical analysis. Visualization of the data was done with the R studio software (version 4.1.2) or GraphPad PRISM (version 8.4.2). Data were deposited on PRIDE (Project Name: Whole cell proteome of U2OS cells or SAOS-2 cells depleted for SENP3; Project accession: PXD037800).

## Bioinformatic tools
**MS data.** The public available ShinyGO tool http://bioinformatics.sdstate.edu/go/ (version 0.76.1), provided by the Bioinformatics Research Group of the South Dakota State University, was used for GO Biological Process as well as KEGG pathway analysis[62–64]. The FDR cutoff was set to 0.05 and a minimum pathway size of 2 as well as a maximum pathway size of 2000 was applied. As background, all protein-coding genes of the human genome were used. The respective pathways were selected by FDR and sorted by fold enrichment. STRING networks were generated using the public available STRING

database (version 11.5) setting the parameters to high confidence and allow experiments and databases as sources. The final visualization was done with the Cytoscape program and Adobe Illustrator. Proteins that were included in the GO- and STRING-analysis were chosen by the following criteria: Interactors need to be enriched with a log2 ratio ≥ 1 and a p value ≥ 1.3. For targets, only proteins with a log2 ratio ≥ 0.58 and a p value ≥ 1.3 were considered.

**RNA-seq data.** For transcript quantification, the GRCh38 reference transcriptome has been used in Salmon (RRID:SCR_017036) or bowtie2 (RRID:SCR_016368)[65]. A count matrix has been generated using htseq-count (RRID:SCR_011867)[66]. After filtering genes with >5 reads in >80% of samples were included in the analysis. Differential gene expression has been performed using DESeq2 (RRID:SCR_015687)[67]. Data dispersion was estimated with a parametric fit using the genotype as an explanatory variable. Wald test was used to determine differentially regulated genes. Subsequently, a reduced log2 fold change was calculated. Genes were judged to be differentially regulated if the nominal p value was <0.05. Regularized log-transformed (rlog) data were used for further downstream analysis. For the gene set enrichment analysis (GSEA) we used GeneTrail3 (RRID:SCR_006250)[68] and gene signatures from the Molecular Signature Database (RRID:SCR_016863)[69].

## Quantification and statistical analysis
Quantification of western blots was done using the LI-COR Image Studio™ software following the manufacturer's instructions. All displayed western blots are representative examples and were validated in at least two independent experiments. Quantification of the northern blots was done using the ImageQuant software. The statistical analysis was done using two-tailed Student's t-tests. FACS data analysis and statistics were done with FlowJo (FlowJo, LLC) and GraphPad Prism (version 8.4.2). If not stated otherwise, all experiments were done with four independent replicates applying unpaired t-tests to determine significance level. Statistics and analysis of the qPCR data was done with Microsoft Excel and GraphPad Prism (version 8.4.2) using unpaired t-testing to verify the significance level. Analysis and statistics of the MS data (IPs and whole-cell proteomes) was done using Perseus software (version 1.6.7.0). In brief, for IP data contaminants, reverse entries and hits only identified by a modified peptide were removed prior to log2 value calculation of the LFQ intensities. Samples were grouped respective to the number of replicates (triplicates for interactomes and SENP3 targetome, four replicates for SENP5 targetome) and the matrix was filtered for minimal two valid values (SENP3 targetome) in at least one group or minimal three valid values (SENP5 targetome) in at least one group. Not matching rows were discarded. Afterward imputation of missing values, based on normal distribution, was done using default settings of Perseus. Finally, Student's t-test was performed by applying a Benjamini–Hochberg FDR of 0.05. Whole-cell proteomes were analyzed as follows: after removal of contaminants, reverse entries and hits only identified by a modified peptide, samples were grouped into four replicates (SAOS-2 siSenp3-1 in triplicate) and the matrix was filtered for minimal 4 valid values in at least one group (U2OS data) or minimal 11 valid values in total (SAOS-2). Log2 transformation of the normalized abundances was done before Student's t-test was performed applying a Benjamini–Hochberg FDR of 0.05. Microsoft Excel was used to determine significant hits by using following criteria: log2 ratio ≥ 1, −log10 p value ≥ 1.3 (interactomes), log2 ratio ≥ 0.58, −log10 p value ≥ 1.3 (targetomes and proteomes).

The quantification of the fluorescent intensities of the microscopic images was done with the CellProfiler4.2.1 software and statistics was calculated in GraphPad Prism 5 for Mac OS X[70,71]. Masking of the cytoplasm, nuclei or nucleoli was done using the cytoplasmic Halo signal of the "old" ribosomes, the DAPI signal or the BMS1 signal, respectively. The Ribo-Halo experiments were performed as four replicates and two-sided t-testing was used to compare the mean ratios

of each replicate. The UTP14 localization experiments were performed as triplicates and an unpaired t-test was performed to compare the mean ratios of each replicate, which were additionally normalized to mean nuclear intensity of the UTP14 signal.

## Reporting summary

Further information on research design is available in the Nature Portfolio Reporting Summary linked to this article.

## Data availability

The MS data generated in this study have been deposited in the PRIDE database under the following project names and accession numbers. Project Name: Identification of SENP3 and SENP5 target proteins by endogenous SUMO2/3 IP-MS; accession: PXD037793, PXD037796. Project Name: Identification of UTP14A interactors by endogenous UTP14A-IP-MS; accession: PXD043556. Project Name: Whole cell proteome of U2OS cells or SAOS-2 cells depleted for SENP3; accession: PXD037800. The RNA-seq data generated in this study have been deposited in the European Nucleotide Archive and are accessible via accession ID: PRJEB57219. Source Data for all MS and RNA-seq experiments are provided with this paper as Supplementary data.

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

## Acknowledgements

This work was supported by Deutsche Forschungsgemeinschaft (DFG grants MU-1764/6-Project ID:465470262 and MU-1764/7-Project ID:494535244) and by the Clusterproject ENABLE funded by the Hessian Ministry for Science and the Arts to S.M., DFG grants KE 222/10-1 and KE 222/11-1 to U.K., SFB1565 (Project ID:469281184) to K.E.B.; CPI, EXC 2026, Project ID: 390649896 and SFB 1177 Project ID: 259130777 to M.K., DFG grant WI 6148/1-1-Project ID-529255113 to M.W., Deutsche Krebshilfe (grants 70114425 and 70114724 to U.K., grant 70114823 to S.M., grant 70115444 to U.K. and M.W.). We thank all members of the Quantitative Proteomics Unit at IBC2 (Goethe University, Frankfurt), in particular Martin Adrian-Allgood for technical help, Kristina Wagner for preparing LC columns and David Krause for help in (bio)informatics. Additionally, we thank Mona Honemann-Capito for technical assistance. We thank the DFG for funding the LC-MS systems used in this study: Orbitrap Fusion LUMOS: FuGG Project ID: 403765277; Q Exactive HF: Project ID: 259130777, SFB 1177 – Selective Autophagy.

## Author contributions

S.M. conceptualized the project and supervised J.D., H.M., K.W., P.H., K.S., T.P. and S.H. J.D. performed all experiments except stated otherwise below. H.M. generated the HaloRPL/RPS cell lines and performed experiments shown in Fig. 4c–e and related Supplementary figures. J.D.G. performed the experiments shown in Figs. 3e and 4a, b and related Supplementary figures under supervision of K.E.B. K.W. performed the experiments shown in Fig. 2g and related Supplementary figures as well as Supplementary Fig. 2b. K.W. and K.S. generated the data shown in Fig. 1b. K.S. generated the data shown in Supplementary Fig. 1b, c. P.H. contributed to data shown in Fig. 2b. T.P. and S.H. generated the data shown in Figs. 1a and 2a. S.U. performed the experiments shown in Fig. 8c, d under supervision of M.W. and U.K. M.W. supported data analysis of the RNA-Seq data. M.K. provided parental RPE1 and RPE1^Δp53 cells. G.T. run the mass

spectrometers and supported data analysis. S.M. wrote the manuscript together with J.D. and K.E.B. All authors reviewed the manuscript providing valuable comments.

## Funding

## Competing interests
The authors declare no competing interests.
