## [Peer Review File · Nature Communications]

Characterization of nucleolar SUMO isopeptidases unveils a general p53-independent checkpoint of impaired ribosome biogenesisREVIEWER COMMENTS

Reviewer #1 (Remarks to the Author):

In this manuscript Donig et al., used an unbiased proteomic approach to identify interaction partners and substrates of the SUMO deconjugases SENP3 and SENP5. The data confirms that SENP3 and SENP5 regulates ribosome biogenesis by controlling the SUMOylation status of distinct 60S and 40S accessory factors. They also observed that inactivation of both SENP3 and SENP5 leads to impaired cell cycle progression and the induction of a p53-independent checkpoint that may rely in part on the downregulation of CDK6. They also propose that downregulation of CDK6 is a general mechanism triggered upon inhibition of ribosome biogenesis, regardless of p53 status. Inhibition of ribosome biogenesis has in recent years emerged as a new therapeutic approach for cancer. Thus, elucidating signaling pathways involved in regulating cell proliferation after ribosome biogenesis inhibition is of clinical importance. A large majority of cancers are p53-mutated, making it critical to understand the effects of inhibiting ribosome synthesis in p53-WT vs. p53-mutant cancer cells. The data presented here is interesting as it identifies several new potential substrates for the largely uncharacterized SENP5 SUMO deconjugase. Additionally, the data suggests that downregulation of CDK6 may be important to establish a p53-independent checkpoint cell cycle arrest resulting from impaired ribosome biogenesis. Although there are several new discoveries, the manuscript presents two different stories (Identification of new SENP5 substrates and, evidence of a p53-independent ribosome biogenesis checkpoint) that should each be further explored and developed in terms of mechanisms to solidify the findings and conclusions. In general, re-expression rescue experiments of SENP3 and 5 in RNAi treated cells are missing and should be provided for key experiments.

Specific comments:

1. For the FLAG IP MS, the authors should provide WB showing the levels of SENP3CAT and SENP5CAT expressed. Transient expression in HEK cells can lead to extremely high levels of expression. How do the ectopic levels compare to endogenous levels? Are some of the interactions found to be the result of forced overexpression as several the proteins identified do not seem to be substrates of deSUMOylation. Do these CAT mutants localize to the nucleolus? IF data showing the proper localization of the FLAG constructs should be added.

2. The authors should add more details on how the SENP3 KO cells were generated: - sequences of the guide RNA utilized, explain whether clones or pools of infected cells were used, provide WB showing absence of full length or truncated SENP3 protein. Due to the random nature of insertions/deletions it would be important to show that a clean deletion background and a homogenous pool of cells was used for the experiment.

3. Referring to the anti-SUMO2/3 affinity purification in SENP3 KO cells, the authors state "These changes are not simply due to alterations in protein expression as confirmed by proteome analysis of control cells and cells depleted from SENP3 (Supplementary table VIII). I do not see any supp table VIII. Perhaps they mean supp table VII? If this is the case, Table VII present data on a different cell line suing

RNAi of SENP3 and the KO used in the pull-down experiment. The authors should show matched input sample data collected from the same KO cells prior to pull-down as there may be differences between depletion by RNAi vs. the KO cells.

4. The proteomic analysis has identified a significant number of potential SENP5 interacting partners and targets. However, this trove of interesting data is only superficially looked at, and there is an overall lack of mechanistic exploration. SENP5 (and or SENP3) connection to the SSU node should be further explored to better understand their roles in ribosome synthesis. The interaction with UTP14 should be solidified by performing co-IP experiments of endogenous SENP3 and SENP5 as well as reverse IP of UTP14. What happens to UTP14 in absence of SENP3/SENP5? Does SUMOylation prevent its recruitment to the SSU processome? Please provide re-expression of SENP3 and 5 WT and CAT in the RNAi treated cells for fig 4G.

5. The authors conclude that SENP5 does not catalyse deSUMOylation at pre-60S particles. However, the data presented show that SENP5 interacts with ribosomal proteins of the large sub-unit and co-sediments with 60S particles (although less predominantly than with the 40S). The authors should explore the effects of SENP5 KD on rRNA synthesis or processing and look at pre-40 and 60S biogenesis by performing sucrose gradient fractionations. This would support the Halo-tag data that is more or less convincing (see below).

6. Although structural modelling of RPL28-Halo supports proper integration of the fusion protein into ribosomes, it should be experimentally tested. There is a lot less RPS3- and RPL28-Halo protein expressed compared to endogenous (supp fig 1B). Are the Halo-tag clones pure heterozygous populations? This could suggest a problem in stability or incorporation into pre-ribosomes. It's not clear on the images provided in Fig 3C what is green, yellow or red. Controls of single and double labelling should be provided. Does the nucleolar accumulation of green RPL28 (even in the control cells) suggest the tagged version may affect 60S export? Data for SENP5 KD alone for both RPS3- and RPL28-Halo is missing. Please provide quantification for this analysis.

7. Fig 4G. There are inconsistencies in the results obtained with the different SENP3 siRNAs. EX: siSENP3-2 increases the G1 fractions whereas siSENP3-1 does not. Instead, this siRNA increases the G2 population. This suggests that some phenotypes observed could be due to off target effects and thus, re-expression rescue experiments should be provided.

8. Fig 5; It is difficult to reconcile why depletion of SENP3 leads to a G1 arrest in p53 depleted cells and G2 impaired progression in p53 WT cells. SENP3 and SENP5 have been shown to be required for proper mitotic spindle assembly and chromosome segregation where they regulate the activity of Aurora A. Thus, the cell cycle effects observed could be linked to these functions and not necessarily regulation of ribosome biogenesis. The authors should demonstrate this is not the case and further discuss in the interpretation of their results.

9. The authors should verify the upregulation of the CKI p16, which could be responsible for the cell cycle arrest (and down regulation of CDK6) induced by ribosome synthesis inhibition in p53-WT and deficient cells.

10. CDK4 and CDK6 are thought to play interchangeable roles in cell cycle regulation. In absence of CDK6, CDK4 is thought to be able to compensate. Is CDK6 the predominant G1 CDK expressed in the cell lines tested? The authors should demonstrate that down-regulation of CDK6 is critical to mediate the p53-independent checkpoint arrest. If indeed CDK6 is downregulated transcriptionally after inhibition of ribosome biogenesis, ectopic expression of CDK6 and/or a constitutively active mutant should reverse the impaired cell cycle progression in both p53 WT and null cells.

11. Does depletion of SENP3 and SENP5 in p53-mutated cancer cells (not p53-null cells) also leads to cell cycle arrest and down-regulation of CDK6? As the vast majority of cancers are not p53 null but instead harbor p53 mutations, it would be more clinically relevant to determine whether these observations hold true in such cancer cells.

Reviewer #2 (Remarks to the Author):

The authors study the roles of SENP3 and SENP5 in ribosome maturation using proteomics. They identify potential SUMO substrates for these proteases without properly investigating how sumoylation regulates their function. Subsequently, they identify a decrease in CDK6 upon SENP3 and SENP5 knockdown that is independent of p53, but the molecular mechanism underlying this decrease in CDK6 remains unclear. Whereas the findings are interesting, the authors need to fill several gaps in the story as outlined below.

Main comments

1. Overall, the paper is descriptive and lacks a proper molecular mechanism to explain how SENP3 and SENP5 regulate ribosomal maturation as acknowledged by the authors in the discussion. The molecular mechanism underlying regulation of ribosome maturation by SENP3/5 is unclear. What are the most critical SUMO substrates regulated by SENP3/5? How are they regulated by sumoylation? Sumoylation mutants of these substrates need to be studied in detail to provide proper mechanistic understanding. The authors speculate about the mechanism in the discussion section. It would be important to experimentally address these speculations.

2. Likewise, a proper molecular mechanism is missing to explain how perturbation of ribosome biogenesis affects CDK6 expression at the transcript level. The authors likewise acknowledge this in the discussion section "The detailed signalling processes that connect impaired ribosome biogenesis to CDK6 alterations are currently unknown". Subsequently, they start speculating about potential mechanisms. It would be important to experimentally address these speculations. CDK6 downregulation could be a consequence rather than a cause of the observed changes in cell cycle progression upon SENP3 and SENP5 knockdown.

3. The functional overlap between SENP3 and SENP5 appears to be limited. Please check if you can find

more SUMO substrates besides UTP14A that are regulated by both proteases. Double knockdowns need to be performed and the results need to be compared to the single knockdowns. If the proteases have unique functions for ribosome maturation, then I recommend to split the current manuscript in two separate manuscripts where the function of each SUMO protease is separately addressed and proper molecular mechanisms are provided. If the protease have overlapping functions, then please systematically compare single knockdowns to double knockdowns.

4. Potential overexpression artefacts – what are the expression levels of Flag-SEN3CAT and Flag-SEN5CAT compared to endogenous counterparts? Furthermore, compare interactors of catalytic dead mutants versus wild-type to enable filtering the data since substrates are expected to bind stronger to catalytic dead SENPs.

5. Figures 1 and 2 – What is the explanation for the very limited overlap between interactors and potential SUMO substrates? Are all other substrates regulated in an indirect manner? Are both methods fairly a-specific? It would be important to confirm sets of key substrates by IP -Westerns, not just in total lysate because there is no formal proof that the extra bands represent sumoylation.

6. Figure 3c and d – Include single knockdown of SEN3 in Figure 3d and single knockdown of SEN5 in Figure 3c and d.

7. Figure 3d – Interesting to see that knockdown of UTP14A reduces ribosome maturation. The authors should check whether sumoylation of UTP14A is important by knocking down UTP14A and rescuing with sumoylation-deficient UTP14A.

8. Figure 4 and 5 – The authors should check whether knocking down SEN3 and SEN5 simultaneously further reduces CDK6 and strengthens the phenotype. Knockdown efficiency of SEN5 should be checked by Western.

9. Figure 6 – Interesting to see that the different knockdowns reduce CDK6, but how does SUMO regulate the target proteins? It would be important to perform knockdown and rescue experiments with sumoylation-deficient mutants.

10. Do the findings have pathological relevance? The authors speculate in the discussion about the pathological relevance in AML. It would be important to provide experimental evidence for the pathological relevance of their findings.

Reviewer #3 (Remarks to the Author):

In this Manuscript, the authors present a study on the role of nucleolar SUMO isopeptidases during ribosome synthesis. In the course of this study, the authors also discovered a new control point of altered ribosome biogenesis, independent of p53. Indeed, in the course of their work, the authors identified a role for SEN3 in the synthesis of the two ribosomal subunits and also a rather specific role

for SENP5 in the assembly of the 40S subunit. Second, they identified that depletion of these factors promoted G1 arrest during the cell cycle, a known consequence of defects in ribosome synthesis. This specific response to disruption of ribosome synthesis, also known as IRBC (impaired ribosome biogenesis check-point), is mediated by ribosomal proteins released as a free pool that are able to sequester Mdm2 and promote p53 stabilization. As a marker of IRBC activation, the authors noted an increase in the expression of p53 pathway components (Figure 4). In addition to this pathway, the authors noted a clear reduction in CDK6 protein following SENP3 depletion. To test whether this reduction in CDK6 was specific to SENP3/SENP5 depletion, they also tested the effect of other ribosomal assembly factors and drug on CDK6 expression, and found that CDK6 was a novel general response independent of p53 following ribosomal stress.

All data presented are of good quality and support a function of SENP3/SENP5 in ribosome assembly as well as a specific effect of ribosome assembly defects on CDK6 expression that could explain the p53-independent response to ribosomal stress. This study is very compelling and reveals a novel p53-independent CDK6 pathway activated after ribosomal disruption, a pathway that is widely targeted by chemotherapeutic agents. This will undoubtedly benefit to the field.

Here are my comments:

1- At the beginning of the article, the authors characterize the role of SENP3 and SENP5 on the modification of two ribosome assembly factors: LAS1L and UTP14A involved in pre-60S and SSU processome processing respectively. The authors then propose that SENP3 is involved in 40S and 60S processing while SENP5 only has a function in small ribosomal subunit processing. This function is redundant with that of SENP3 in the same pathway. This is confirmed by sucrose sedimentation assays in which SENP3 co-migrates with the 40S and 60S peaks while SENP5 is mainly present in the 40S fractions. Next, to fully assess the independent role of SENP3 and SENP5, the authors set out to use a rather robust and elegant assay: the Ribo-Halo assays (Figure 3C). From my perspective, two conditions are missing in this test and could improve the outcome:

a- A check for no effect of SENP5 depletion on RPL28-halo as a negative control, to confirm that SENP5 has no role on 60S maturation and that the effect of SENP3 is not due to its role on the sumoylation of UTP14a.

b- An independent depletion assay with separate depletion of SENP3 and SENP5 on RPS3-Halo, to fully assess their independent function during small ribosomal subunit processing.

2- Regarding the effect of CDK6 loss on ribosome biogenesis, the authors suggest in their last results section and discussion that CDK6 depletion could explain part of the G1/S arrest observed in p53-deficient cells following the disruption of ribosome synthesis. To fully support this role, it would be important to overproduce CDK6 in both p53-deficient and p53-sufficient cells and to analyze the effect of ribosome synthesis disruption (by CX-5461 for example) on the cell cycle. One would expect to reduce G1/S arrest in such cells compare to control cells that do not overexposes CDK6.

Point-by-point answer to reviewer comments

Reviewer #1 (Remarks to the Author):

In this manuscript Donig et al., used an unbiased proteomic approach to identify interaction partners and substrates of the SUMO deconjugases SENP3 and SENP5. The data confirms that SENP3 and SENP5 regulates ribosome biogenesis by controlling the SUMOylation status of distinct 60S and 40S accessory factors. They also observed that inactivation of both SENP3 and SENP5 leads to impaired cell cycle progression and the induction of a p53-independent checkpoint that may rely in part on the downregulation of CDK6. They also propose that downregulation of CDK6 is a general mechanism triggered upon inhibition of ribosome biogenesis, regardless of p53 status. Inhibition of ribosome biogenesis has in recent years emerged as a new therapeutic approach for cancer. Thus, elucidating signaling pathways involved in regulating cell proliferation after ribosome biogenesis inhibition is of clinical importance. A large majority of cancers are p53-mutated, making it critical to understand the effects of inhibiting ribosome synthesis in p53-WT vs. p53-mutant cancer cells. The data presented here is interesting as it identifies several new potential substrates for the largely uncharacterized SENP5 SUMO deconjugase. Additionally, the data suggests that downregulation of CDK6 may be important to establish a p53-independent checkpoint cell cycle arrest resulting from impaired ribosome biogenesis. Although there are several new discoveries, the manuscript presents two different stories (Identification of new SENP5 substrates and, evidence of a p53-independent ribosome biogenesis checkpoint) that should each be further explored and developed in terms of mechanisms to solidify the findings and conclusions. In general, re-expression rescue experiments of SENP3 and 5 in RNAi treated cells are missing and should be provided for key experiments.

We thank the reviewer for their overall positive view on our work. We appreciate the constructive criticism and input. In the following we will address the concerns point-by-point.

Specific comments:

1. For the FLAG IP MS, the authors should provide WB showing the levels of SENP3CAT and SENP5CAT expressed. Transient expression in HEK cells can lead to extremely high levels of expression. How do the ectopic levels compare to endogenous levels? Are some of the interactions found to be the result of forced overexpression as several the proteins identified do not seem to be substrates of deSUMOylation. Do these CAT mutants localize to the nucleolus? IF data showing the proper localization of the FLAG constructs should be added.

Following the reviewer's suggestion we have added IF data demonstrating proper nucleolar localization of the Flag-tagged catalytic mutants of SENP3 and SENP5 (new Supplementary Figure 1B). Furthermore, we added a Figure showing expression of Flag-SENP3/5 in comparison to the respective endogenous proteins (new Supplementary Figure 1A). As expected, and anticipated by the reviewer, the expression exceeds endogenous protein levels. Although we cannot exclude that this can occasionally result in unspecific interactions, the observation that only a subset of the co-immunoprecipitated proteins are substrates of the respective SENPs is more likely explained by the fact that anti-Flag-SENP3/SENP5 IPs not only capture direct interactors, but enrich for protein complexes, where not all individual subunits are targets of SENPs. We would also like to stress that in the MS experiment most Flag-SENP3/5 interactions (including SENP5-UTP14A association) were only detected with the catalytic mutants, but not with the respective wild-type proteins (even though they were

expressed at a similar level). The data for IPs with WT Flag-SENP3/5 (that was generated in parallel to the IPs with the catalytic mutants) is now included in Supplementary table I. This finding is in line with the general view in the field of Ub/Ubl deconjugases that catalytic mutants of deconjugating enzymes act as substrate traps. In IPs the catalytic mutants not only trap substrates, but also proteins associated with these substrates.

2. The authors should add more details on how the SENP3 KO cells were generated: - sequences of the guide RNA utilized, explain whether clones or pools of infected cells were used, provide WB showing absence of full length or truncated SENP3 protein. Due to the random nature of insertions/deletions it would be important to show that a clean deletion background and a homogenous pool of cells was used for the experiment.

Following the reviewer's suggestion we provide more information concerning the SENP3 KO cells in the methods section of our manuscript. The following guide RNAs were used to create SENP3 KO cells (U2OS):

guide 1 forward gggctccttactctgtacgc
guide 1 reverse gcgtacagagtaaggagccc
guide 2 forward cctccacctgacttgagtgcg
guide 2 reverse cgactcaagtcaggtggagg
guide 3 forward cagcaatgtgtgcagcatcg
guide 3 reverse cgatgctgcacacattgctg

The guides were designed with the sgRNA selection tool of the Broad Institute (2016 edition). For virus production in HEK293T cells, all sgRNAs were pooled to target SENP3. After transduction of the target cells (U2OS) success of the KO was validated by immunoblotting (Supplementary Figure 1C). Experiments were done using pooled KO cells. The method section (CRISPR methods) was updated with all mentioned details.

3. Referring to the anti-SUMO2/3 affinity purification in SENP3 KO cells, the authors state "These changes are not simply due to alterations in protein expression as confirmed by proteome analysis of control cells and cells depleted from SENP3 (Supplementary table VIII). I do not see any supp table VIII. Perhaps they mean supp table VII? If this is the case, Table VII present data on a different cell line using RNAi of SENP3 and the KO used in the pull-down experiment. The authors should show matched input sample data collected from the same KO cells prior to pull-down as there may be differences between depletion by RNAi vs. the KO cells.

We thank the reviewer for making us aware of this confusion. The numbering of the respective table was incorrect. We have now updated the numbering of all supplementary tables and changed it in the manuscript. The table providing information concerning the proteomes of the SENP3 KO cells used for identification of SENP3 target proteins is now "Supplementary table III_Proteome for SENP3 targets".

4. The proteomic analysis has identified a significant number of potential SENP5 interacting partners and targets. However, this trove of interesting data is only superficially looked at, and there is an overall lack of mechanistic exploration. SENP5 (and or SENP3) connection to the SSU node should be further explored to better understand their roles in ribosome synthesis. The interaction with UTP14 should be solidified by performing co-IP experiments of endogenous SENP3 and SENP5 as well as reverse IP of UTP14. What happens to UTP14 in absence of SENP3/SENP5? Does SUMOylation prevents it recruitment to the SSU processome? Please provide re-expression of SENP3 and 5 WT and CAT in the RNAi treated cells for fig 4G.

We appreciate the reviewer's comment that the dataset of SENP5 interactors and targets is interesting. We also acknowledge that in our initial version we did not further explore the

SEN3/5 connection to the SSU/pre-40S node and the functional consequence of persistent UTP14A SUMOylation. To address these specific points, we have now added new datasets and experiments.

The interaction with UTP14 should be solidified by performing co-IP experiments of endogenous SEN3 and SEN5 as well as reverse IP of UTP14.

We agree that endogenous IPs are considered as a gold standard for demonstrating protein-protein interactions in cells. However, the interactions of Ub/Ubl deconjugation enzymes with their respective substrates are in most cases very transient precluding their detection by endogenous IPs. (A notable exception is the very robust association of SEN3 with the rixosome complex). As mentioned above, for that reason expression of catalytic mutants is often the only way to capture substrates.

What happens to UTP14 in absence of SEN3/SEN5? Does SUMOylation prevents it recruitment to the SSU processome?

We agree with the reviewer that these are key questions to understand the molecular mechanisms that link SUMOylation of UTP14A to the control of ribosome biogenesis. To address this question, we performed an endogenous immunoprecipitation experiment of UTP14A in control cells and cells depleted from SEN3 and SEN5 individually or in combination and analysed the immunoprecipitated material by mass-spectrometry (new Supplementary Table VI). In control cells, 29 proteins were at least 2-fold enriched in UTP14A IPs compared to IgG controls (new Figure 3A). STRING and GO term analysis of these proteins reveals a strong enrichment of an interconnected network of RPS proteins and 40S biogenesis factors (new Figure 3B, C). Upon individual depletion of either SEN3 or SEN5 or co-depletion of both proteins, a set of RPS proteins and 40S trans-acting factors, including RRP12, TSR1 and NOP14, exhibit reduced association of UTP14A (new Figure 3D, Supplementary Table VI). Proteome analysis confirmed that this is not due to alterations in expression of the respective proteins (new Supplementary Table VII). We therefore hypothesized that the association of UTP14A with pre-40S particles is impaired upon depletion of SEN3/5. To validate this point, we performed sucrose gradient sedimentation assays following depletion of SEN3/5. In control cells, UTP14A co-fractionates with RPS3A in pre-40S particles (fraction 5-8) (new Figure 3E). Intriguingly, however, upon depletion of SEN3/5 the amount of UTP14A is reduced in these fractions supporting the idea that unrestricted SUMOylation impairs the association of UTP14A with pre-40S/SSU processome particles. In agreement with these data, UTP14A is shifted from the nucleolus to the nucleoplasmic fraction in the absence of either SEN3 or SEN5 (new Supplementary Figure 2A). MS datasets identified K733 as the major SUMOylation site in UTP14A. Based on the available cryo-EM structures we hypothesize that in case of UTP14A, the attachment of SUMO at K733 sterically hinders the incorporation of UTP14A in pre-ribosomes (new Supplementary Figure 2B). Notably, in contrast to UTP14A, the association of PELP1 with pre-60S particles was enhanced upon SEN3/5 depletion supporting our previous concept of rixosome trapping at pre-60S particles upon constitutive SUMOylation. Altogether, these data indicate that dynamics of SUMO conjugation-deconjugation shape pre-ribosomal particles at specific stages of the maturation pathway.

Please provide re-expression of SEN3 and 5 WT and CAT in the RNAi treated cells for fig 4G.

We appreciate the reviewer's comment and suggestion and accordingly tried to address this point. To this end, we generated a U2OS cell line expressing SEN3 under a doxycycline-inducible promoter. When monitoring CDK6 level by immunoblotting upon induced expression of SEN3, we found, that not only SEN3 KD, but also enhanced expression of SEN3 reduces the protein level of CDK6 (see below). These data demonstrate that cells do not

tolerate an imbalance of SENP3 expression and SENP3-mediated deSUMOylation. In agreement with our data on SENP3-mediated control of the rixosome, we conclude that both depletion or overexpression of SENP3 will affect ribosome maturation and therefore activate the identified CDK6-dependent checkpoint. Even though this observation precludes knock-down re-expression experiments, we feel that these data strengthen the concept of altered SENP3-signaling impinging on CDK6 expression.

Nevertheless, to rule out any possible off target effects of siRNA mediated SENP3 KD we tested additional SENP3 siRNAs (4 in total) and confirmed the observed phenotype (CDK6 loss) (new Figure 6D).

5. The authors conclude that SENP5 does not catalyse deSUMOylation at pre-60S particles. However, the data presented show that SENP5 interacts with ribosomal proteins of the large sub-unit and co-sediments with 60S particles (although less predominantly than with the 40S). The authors should explore the effects of SENP5 KD on rRNA synthesis or processing and look at pre-40 and 60S biogenesis by performing sucrose gradient fractionations. This would support the Halo-tag data that is more or less convincing (see below).

Following the reviewer's suggestion, we performed northern blots using specific probes hybridizing within the internal transcribed spacers ITS1 and ITS2 that detect all major pre-rRNA processing intermediates (new Supplementary Figure 3A). In accordance with our published findings, depletion of SENP3 results in the accumulation of the 32S precursor demonstrating the involvement of SENP3 in conversion of this pre-rRNA to the mature 28S rRNA of the 60S subunit (new Figure 4B and Supplementary Figure 3A). In line with the findings that SENP5 does not interact with or target the rixosome components or other pre-60S biogenesis factors, depletion of SENP5 does not affect 32S levels. By contrast, depletion of either SENP5 or SENP3 reduces the level of the 18SE rRNA species (new Figure 4B and Supplementary Figure 3A), which represents a direct precursor of the 18S rRNA of the 40S subunit, indicating that both nucleolar isopeptidases are important for small subunit maturation.

This would support the Halo-tag data that is more or less convincing (see below).

In line with the reviewer's comment, the results from northern blots support the data obtained in Halo-assays. We take note of the reviewer's criticisms on the Ribo-Halo assay and accordingly repeated the assays and quantified the data from single and combined depletion of SENP3/5 (see below, point 6).

6. Although structural modelling of RPL28-Halo supports proper integration of the fusion protein into ribosomes, it should be experimentally tested. There is a lot less RPS3- and RPL28-Halo protein expressed compared to endogenous (supp fig 1B). Are the Halo-tag clones pure heterozygous populations? This could suggest a problem in stability or incorporation into pre-ribosomes. It's not clear on the images provided in Fig 3C what is green, yellow or red. Controls of single and double labelling should be provided. Does the nucleolar accumulation of green RPL28 (even in the control cells) suggests the tagged version may affect 60S export? Data for SENP5 KD alone for both RPS3- and RPL28-Halo is missing. Please provide quantification for this analysis.

We thank the reviewer for raising this point. HeLa cells harbour multiple alleles encoding RPS3 and RPL28. As mentioned by the reviewer, we detect both untagged endogenous RPS2/RL28 as well as the Halo-tagged proteins indicating that endogenous tagging did not occur on all alleles (Supplementary Figure 3B, previous Supplementary Figure 1B). This explains the differences in proteins levels. Notably, the tagging of only some alleles does not affect the assay system, since we only follow a subset of the tagged ribosomes. To make sure that RPS3- and RPL28-Halo are properly incorporated into pre-ribosomes we now performed sucrose gradient assays. The data support proper integration of both fusion proteins into ribosomes, also validating published data for RPS3-Halo (new Supplementary Figure 3C and 3D).

It's not clear on the images provided in Fig 3C what is green, yellow or red.

We apologize for not properly explaining the labeling of cells in the Ribo-Halo assays. We now try to clarify this point by explaining the labeling procedure and the workflow in more detail. Furthermore, we repeated the experiments including single and combined SENP3/5 depletions and quantified the data. Cells expressing the Halo-fused proteins were sequentially labelled with two distinct fluorescent ligands to follow the fate of existing and newly synthesized 40S or 60S ribosomal subunits by immunofluorescence (Figure 4C). To monitor "old" (pre-existing) or nascent 40S/60S ribosomal subunits, we performed sequential labelling with individual red (tetramethylrhodamine; TMR) or green (R110) fluorescent ligands. Imaging of the control RPL28-Halo cells showed a predominant yellow signal resulting from a merge of both "old/red" and newly synthesized/green ribosomal subunits (new Figure 4D). By contrast, upon siRNA-mediated depletion of the pre-60S maturation factor MDN1, the red signal of the old ribosomes was largely dominating in the cytosol, indicating that synthesis of 60S ribosomal subunit was impaired (new Figure 4D, E). A very similar phenotype was observed upon depletion of SENP3, confirming that SENP3 is indeed required for proper 60S maturation (new Figure 4D, E). Consistent with the data from northern blotting analysis of the pre-rRNA processing, depletion of SENP5 did not impair 60S biogenesis (new Figure 4D, E). To see how SENP3/5 affect pre-40S maturation, analogous experiments were performed in the RPS3-Halo cell line. Control cells again showed the presence of both "old" and newly synthesized 40S subunits as indicated by the pre-dominantly yellow (merge "old" and "new" ribosomes) signal in the cytosol (new Figure 4D). Depletion of UTP14A, which was used as a control, shifts the balance towards a red signal indicative of impaired de novo synthesis of 40S subunits (new Figure 4D, E). A comparable scenario was detected when either SENP3 or SENP5 were individually depleted or co-depleted from RPS3-Halo cells (new Figure 4D, E and Supplementary Figure 3E). Quantification of the signals demonstrates that lack of SENP3 has a stronger impact on 40S biogenesis than the lack of SENP5 (new Figure 4E). Importantly, co-depletion of both isopetidases further aggravates the maturation defect (new Supplementary Figure 3E). Altogether, these data demonstrate that nucleolar SENPs are important regulators of ribosome biogenesis with SENP3 controlling both 40 and 60S maturation and SENP5 exhibiting a more specific role in 40S maturation.

Controls of single and double labelling should be provided. Does the nucleolar accumulation of green RPL28 (even in the control cells) suggests the tagged version may affect 60S export? Data for SENP5 KD alone for both RPS3- and RPL28-Halo is missing. Please provide quantification for this analysis.

As suggested by the reviewer, we have added data on individual depletion of SENP3 and SENP5 (new Figure 4D). Furthermore, we have quantified the data (new Figure 4E; Supplementary Figure 3E).

Does the nucleolar accumulation of green RPL28 (even in the control cells) suggests the tagged version may affect 60S export?

RPL28 is incorporated at an early stage in pre-ribosomes explaining its accumulation in the nucleolus when early steps of RiBi are blocked. However, in the new RPL-28 Halo experiments that were included in the MS (new Figure 4E) the nucleolar localization was less pronounced under both impaired RiBi or control conditions. For that reason, we deleted this statement in the revised version.

7. Fig 4G. There are inconsistencies in the results obtained with the different SENP3 siRNAs. EX: siSENP3-2 increases the G1 fractions whereas siSENP3-1 does not. Instead, this siRNA increases the G2 population. This suggests that some phenotypes observed could be due to off target effects and thus, re-expression rescue experiments should be provided.

We thank the reviewer for this valuable comment. We agree that both siRNAs affect cell cycle progression in a slightly different way. To our opinion, this can be explained by the extend of SENP3 depletion, which in turn determines the strength of the p53 response and the increase of p21 levels. High p21 levels typically result in both G2/M and G1/S arrest, while lower levels primarily impact on G1/S transition. siSENP3-1 indeed induces p21 more drastically than SENP3-2 (Figure 5A, F) likely explaining the more pronounced increase in the G2 population. We now briefly refer to this point in the result section. Notably, suprainduction of p53 by disruption of 40S and 60S ribosome biogenesis has already been reported to trigger the activation of a G2/M checkpoint (Fumagalli et al., Genes and Dev., 2012).

8. Fig 5; It is difficult to reconcile why depletion of SENP3 leads to a G1 arrest in p53 depleted cells and G2 impaired progression in p53 WT cells. SENP3 and SENP5 have been shown to be required for proper mitotic spindle assembly and chromosome segregation where they regulate the activity of Aurora A. Thus, the cell cycle effects observed could be linked to these functions and not necessarily regulation of ribosome biogenesis. The authors should demonstrate this is not the case and further discuss in the interpretation of their results.

We fully agree that mitotic functions have been ascribed to SENP3 and SENP5. Although we cannot exclude that these effects contribute to the alteration in cell cycle distribution, we actually do think that the differences we observe in p53-proficient vs. deficient cells are fully consistent with our model. In p53-proficient cells, depletion of SENPs will activate the canonical p53-dependent impaired ribosome biogenesis checkpoint and accordingly affect G1/S and G2/M transition (in particular if both 40S and 60S RiBi are impaired, see above Fumagalli et al.), while the G2/M impairment will be lost in p53-deficient cells. The impaired G1/S transition in p53-deficient cells is fully consistent with downregulation of CDK6 and impaired G1/S transition.

9. The authors should verify the upregulation of the CKI p16, which could be responsible for the cell cycle arrest (and down regulation of CDK6) induced by ribosome synthesis inhibition in p53-WT and deficient cells.

We thank the reviewer for this useful suggestion. Immunoblotting of p53 WT or p53 deficient cells (RPE1 WT, RPE1 p53 KO), treated for 72 h with siRNA against SENP3, SBDS or control siRNA, against p16 demonstrates, that there is no up-regulation of this CDK-inhibitor.

10. CDK4 and CDK6 are thought to play interchangeable roles in cell cycle regulation. In absence of CDK6, CDK4 is thought to be able to compensate. Is CDK6 the predominant G1 CDK expressed in the cell lines tested? The authors should demonstrate that down-regulation of CDK6 is critical to mediate the p53-independent checkpoint arrest. If indeed CDK6 is downregulated transcriptionally after inhibition of ribosome biogenesis, ectopic expression of CDK6 and/or a constitutively active mutant should reverse the impaired cell cycle progression in both p53 WT and null cells.

A possible redundancy and compensation of CDK4 and CDK6 is indeed an important issue. To directly address this point, we depleted CDK4 or CDK6 individually or in combination from p53-deficient RPE1 cells and monitored phospho-RB levels. The effects were compared to pRB phosphorylation in the absence of SENP3. The data are included in new Figure 6J. The data demonstrate that individual depletion of either CDK4 or CDK6 reduces phosphorylation of pRB at S807/811 and T826, while co-depletion of both kinases completely abrogated it. This indicates that CDK4 cannot fully compensate lack of CDK6 in RPE1 cells. Importantly, depletion of SENP3 phenocopies CDK6 depletion, while combined CDK4/SENP3 depletion further dampens pRB phosphorylation recapitulating lack of CDK4/6 (new Figure 6J, new Supplementary Figure 6D). Notably, the reduced pRB phosphorylation upon SENP3 loss could be rescued by CDK6 re-expression indicating that impaired cell cycle progression upon depletion of SENP3 is directly linked to CDK6 and alterations in the phosphorylation status of pRB (new Figure 6K, new Supplementary Figure 6E).

11. Does depletion of SENP3 and SENP5 in p53-mutated cancer cells (not p53-null cells) also leads to cell cycle arrest and down-regulation of CDK6? As the vast majority of cancers are not p53 null but instead harbor p53 mutations, it would be more clinically relevant to determine whether these observations hold true in such cancer cells.

We thank the reviewer for pointing out this important issue. To address this point, we used two cellular models of p53-mutated cells that are characterized by high CDK6 levels. We used the pancreatic cancer cell line BxPC-3 as a high CDK6 model system that concomitantly exhibits low CDK4 expression and additionally lacks functional p53 (new Figure 7A). Depletion

of SENP3 from BxPC-3 cells significantly reduces pRB phosphorylation again recapitulating CDK6 depletion (new Figure 7A). Further, depletion of SENP3 or SENP5 from BxPC-3 cells impairs cell cycle progression as indicated by a reduced number of cells in S phase and an increase in G1 cells (new Figure 7B). Co-depletion of SENP3/5 further reduces S phase cells again indicating a synergism of both isopeptidases. To expand on this data, we performed analogous experiments in Ramos cells, a CDK6-dependent, p53-mutant B cell model. Depletion of SENP3 or SENP5 downregulates CDK6 and affects cell proliferation as indicated by a reduced number of S phase cells (new Figure 7C, D). Altogether these data indicate that SENP3/5 represent a potential vulnerability in p53-mutant tumours.

Reviewer #2 (Remarks to the Author):

The authors study the roles of SENP3 and SENP5 in ribosome maturation using proteomics. They identify potential SUMO substrates for these proteases without properly investigating how sumoylation regulates their function. Subsequently, they identify a decrease in CDK6 upon SENP3 and SENP5 knockdown that is independent of p53, but the molecular mechanism underlying this decrease in CDK6 remains unclear. Whereas the findings are interesting, the authors need to fill several gaps in the story as outlined below.

Main comments

1. Overall, the paper is descriptive and lacks a proper molecular mechanism to explain how SENP3 and SENP5 regulate ribosomal maturation as acknowledged by the authors in the discussion. The molecular mechanism underlying regulation of ribosome maturation by SENP3/5 is unclear. What are the most critical SUMO substrates regulated by SENP3/5? How are they regulated by sumoylation? Sumoylation mutants of these substrates need to be studied in detail to provide proper mechanistic understanding. The authors speculate about the mechanism in the discussion section. It would be important to experimentally address these speculations.

We do acknowledge that in the initial version of the manuscript we did not address the molecular mechanism of SENP3/5-mediated control of 40S ribosome biogenesis through SUMOylation of UTP14A in sufficient detail. We now provide a set of experiments explaining the impact of SENP3/5 in regulation both 60S and 40S maturation. I will address the concerns point-by-point.

What are the most critical SUMO substrates regulated by SENP3/5? How are they regulated by sumoylation?

In our published work we already identified PELP1 as one key substrate of SENP3 in 60S maturation. We demonstrated that SENP3 is part of the mammalian PELP1 complex and revealed that this complex is the functional counterpart of the *Saccharomyces cerevisiae* Rix1 complex (Finkbeiner et al., EMBO J., 2011; Raman et al, Mol Cell 2016). The human PELP1 complex - also termed rixosome - comprises the core components PELP1, WDR18, TEX10 and the associated factors MDN1, LAS1L and NOL9. In our previous work, we have shown that balanced SUMO conjugation-deconjugation coordinates the timely association of MDN1 with the PELP1 complex. We established a model, where SUMOylated PELP1 recruits MDN1, while deconjugation by SENP3 is needed for its release. Accordingly, inactivation of SENP3 traps MDN1 on pre-60S particles. Our unbiased proteomics approach now validates the role of SENP3 in countering PELP1 modification, but goes beyond previous findings. The new data provide evidence that SENP3 counters not only the SUMOylation of LAS1L and PELP1, but also limits modification of additional components, including MDN1, TEX10 and NOL9. This is in line with the general concept that the SUMO conjugation and deconjugation machinery acts on entire groups of proteins that are often found in protein complexes. Based on the new data, we hypothesize that the unrestricted SUMOylation of multiple members of the PELP1 complex cooperates to lock the complex on pre-60S particles. We anticipate that this is due to the "glue-like" properties of multi-valent SUMO-SIM (SUMO-interacting motif) contacts between several subunits of the complex. This interpretation is consistent with the enhanced pre-60S association of PELP1 observed here in cells depleted of SENP3/5 (new Figure 3E).

Concerning key substrates at pre-40S particles the ensemble of our data suggests that UTP14A is at least one key substrate. We provide compelling evidence that persistent SUMOylation of UTP14A prevents its proper association with pre-40S particles. This is based on a series of new experimental data. First, we performed an endogenous

immunoprecipitation experiment of UTP14A in control cells and cells depleted from SENP3 and SENP5 individually or in combination and analysed the immunoprecipitated material by mass-spectrometry (new Supplementary Table VI). In control cells, 29 proteins were at least 2-fold enriched in UTP14A IPs compared to IgG controls (new Figure 3A). STRING and GO term analysis of these proteins reveals a strong enrichment of an interconnected network of RPS proteins and 40S biogenesis factors (new Figure 3B, C). Upon individual depletion of either SENP3 or SENP5 or co-depletion of both proteins, a set of RPS proteins and 40S trans-acting factors, including RRP12, TSR1 and NOP14, exhibit reduced association of UTP14A (new Figure 3D, Supplementary Table VI). Proteome analysis confirmed that is not due to alterations in expression of the respective proteins (new Supplementary Table VII). We therefore hypothesized that the association of UTP14A with pre-40S particles is impaired upon depletion of SENP3/5. To validate this point, we performed sucrose gradient sedimentation assays following depletion of SENP3/5. In control cells, UTP14A co-fractionates with RPS3A in pre-40S particles (fraction 5-8) (new Figure 3E). Intriguingly, however, upon depletion of SENP3/5 the amount of UTP14A is reduced in these fractions supporting the idea that unrestricted SUMOylation impairs the association of UTP14A with pre-40S/SSU processome particles. In agreement with these data, UTP14A is shifted from the nucleolus to the nucleoplasmic fraction in the absence of either SENP3 or SENP5 (new Supplementary Figure 2A). MS datasets identified K733 as the major SUMOylation site in UTP14A. Based on the available cryo-EM structures we hypothesize that in case of UTP14A, the attachment of SUMO at K733 sterically hinders the incorporation of UTP14A in pre-ribosomes (new Supplementary Figure 2B). Notably, in contrast to UTP14A, the association of PELP1 with pre-60S particles was enhanced upon SENP3/5 depletion supporting our previous concept of rixosome trapping at pre-60S particles upon constitutive SUMOylation (new Figure 3E). Altogether, these data indicate that dynamics of SUMO conjugation-deconjugation shape pre-ribosomal particles at specific stages of the maturation pathway with PELP1-rixosome and UTP14 being key substrates.

Sumoylation mutants of these substrates need to be studied in detail to provide proper mechanistic understanding.

Using a SUMO-deficient mutant of PELP1 we have already demonstrated that PELP1 is one key substrate of SENP3 in pre-60S maturation (Raman et al., Mol Cell 2016). While a single SUMO site could be identified in PELP1, the situation is more complex in UTP14A. SUMO proteomics by Hendriks et al. has identified up to 23 SUMO attachment sites in UTP14A. Although the data indicate that K733 is the most abundant site, there is a considerable redundancy making it very challenging to generate SUMO-deficient UTP14A. Furthermore, our data suggest that enhanced SUMOylation prevents pre-ribosome association of UTP14A making it unlikely that a SUMO-deficient mutant of UTP14A would *per se* affect ribosome biogenesis. We therefore feel that the above-mentioned endogenous IP of UTP14A in the presence or absence of SENP3/5 is a better way to address the consequence of persistent UTP14A SUMOylation.

2. Likewise, a proper molecular mechanism is missing to explain how perturbation of ribosome biogenesis affects CDK6 expression at the transcript level. The authors likewise acknowledge this in the discussion section “The detailed signalling processes that connect impaired ribosome biogenesis to CDK6 alterations are currently unknown”. Subsequently, they start speculating about potential mechanisms. It would be important to experimentally address these speculations. CDK6 downregulation could be a consequence rather than a cause of the observed changes in cell cycle progression upon SENP3 and SENP5 knockdown.

We agree that in the initial version of our manuscript we were not yet exploring the molecular events that connect impaired ribosome biogenesis to CDK6 expression and cell cycle control in detail. In an effort to fill this gap we now demonstrate that the major downstream target of

this signaling process is indeed pRB. We show that alterations in CDK6 levels upon SENP3/5 reduction perfectly correlate with phosphorylation of pRB at S807/811 and T826. The data are included in new Figure 6J. The data demonstrate that depletion of SENP3 phenocopies CDK6 depletion (new Figure 6J, new Supplementary Figure 6D). Importantly, the reduced pRB phosphorylation upon SENP3 loss could be rescued by CDK6 re-expression indicating that impaired cell cycle progression upon depletion of SENP3 is directly linked to CDK6 and alterations in the phosphorylation status of pRB (new Figure 6K, new Supplementary Figure 6E).

To support our hypothesis that there is a signaling pathway from ribosomal proteins to the CDK6-pRb axis we depleted a series of RPL and RPS proteins. We anticipated that unassembled ribosomal proteins impinge on CDK6 regulation. In line with this idea, we show that an imbalance of ribosomal proteins generated by depletion of a set of ribosomal proteins downregulates CDK6 (new Figure 8D, Supplementary Figure 7D). Notably, however, depletion of RPL11 or RPS7 does not affect CDK6 possibly indicating their involvement in signaling impaired ribosome biogenesis to the CDK6-pRB axis (Supplementary Figure 7D). Intriguingly, RPL11 has already been proposed to be involved in p53-independent signaling of impaired ribosome biogenesis through its binding to E2F as discussed in detail in the discussion section. Further, RPL11 also recruits miR-24 and the microRNA-induced silencing complex (miRISC) to the MYC RNA transcript regulating its turnover to repress MYC expression in response to ribosomal stress. Our data therefore suggest that RPL11 (potentially together with RPS7) is one critical mediator in the SENP3/RiBi-CDK6 signalling axis. We will explore details of this signalling process in future work, but feel that the additional data already provide important first mechanistic insight into the newly identified pathway.

3. The functional overlap between SENP3 and SENP5 appears to be limited. Please check if you can find more SUMO substrates besides UTP14A that are regulated by both proteases. Double knockdowns need to be performed and the results need to be compared to the single knockdowns. If the proteases have unique functions for ribosome maturation, then I recommend to split the current manuscript in two separate manuscripts where the function of each SUMO protease is separately addressed and proper molecular mechanisms are provided. If the protease have overlapping functions, then please systematically compare single knockdowns to double knockdowns.

To more clearly dissect distinct roles of SENP3 and SENP5 in 40S/60S RiBi we now have performed Northern Blot experiments to monitor rRNA processing. Using specific probes (within ITS1 and ITS2) that detect distinct rRNA intermediates we demonstrate that depletion of SENP3 results in the accumulation of the 32S precursor confirming our published work on the involvement of SENP3 in 32S to 28S rRNA processing (new Figure 4B and Suppl. Figure 3A). Noteworthy, depletion of SENP5 does not affect 28S maturation. However, depletion of either SENP5 or SENP3 reduces the level of the 18SE rRNA species (new Figure 4B and Suppl. Figure 3A), which represents a direct precursor of the 18S rRNA, indicating that both nucleolar isopeptidases are critical for small subunit maturation. These data are perfectly in line with the results from the HaLo assays, where we also demonstrate a regulatory role SENP3 - but not SENP5 - in large subunit biogenesis (new Figure 4D). By contrast both isopeptidases synergize in controlling small subunit maturation as demonstrated by the aggravated 40S maturation defect in co-depletion experiments (Supplementary Figure 3E). Following the reviewer's suggestion we have now added a series of experiments, in which we co-depleted SENP3 and SENP5. In various other assays co-depletion of both isopeptidases strengthens the phenotyp (see below).

At the level of individual substrates SENP3 and SENP5 also show clear differences. While SENP3 can catalyze demodification of both 60S and 40S trans-acting factors, SENP5 activity is limited to 40S regulators thereby fully supporting the functional data.

The redundant role of SENP3/5 is best exemplified on UTP14A, where co-depletion of SENP3/5 strongly enhances SUMOylation when compared to individual depletion (new Figure 2G). Similarly, UTP18 and BMS1 exhibit enhanced SUMO2/3 conjugation upon depletion of SENP3 and further enhancement upon co-depletion of SENP3/5, albeit at a far lower level (new Supplementary Figure 1E). These data indicate that UTP14A is indeed a potential prime substrate of SENP3/5 and further suggest that both isopeptidases exert partially redundant functions as deconjugases of UTP14A and potentially other pre-40S regulators. This contrasts with the activities of SENP3 and SENP5 towards the ribosome complex, where loss of SENP5 alone or in combination with SENP3 does not affect the modification of LAS1L indicating that SENP5 does not catalyze deSUMOylation at pre-60S particles (Figure 2G, right panel). Altogether, our datasets strengthen the notion that SENP3 and SENP5 are involved in ribosome biogenesis by controlling the SUMOylation status of distinct trans-acting factors.

4. Potential overexpression artefacts – what are the expression levels of Flag-SENP3CAT and Flag-SENP5CAT compared to endogenous counterparts? Furthermore, compare interactors of catalytic dead mutants versus wild-type to enable filtering the data since substrates are expected to bind stronger to catalytic dead SENPs.

Following the reviewer's suggestion, we added a Figure showing expression of Flag-SENP3/5 in comparison to the respective endogenous proteins (new Supplementary Figure 1A). As expected, the expression exceeds endogenous protein levels. Although we cannot exclude that this causes some unspecific interactions, it is worth noting, that most Flag-SENP3/5 interactions (including SENP5-UTP14A binding) were only detected with the catalytic mutants, but not with the respective wild-type proteins, which were expressed at a similar level. As suggested by the reviewer, the data for IPs with WT Flag-SENP3/5 (that was generated in parallel to the IPs with the catalytic mutants) is now included in Supplementary table I. As mentioned by the reviewer, this finding is in line with the general view that catalytic mutants of deconjugating enzymes act as substrate traps.

5. Figures 1 and 2 – What is the explanation for the very limited overlap between interactors and potential SUMO substrates? Are all other substrates regulated in an indirect manner? Are both methods fairly a-specific?

The observation that only a subset of the interacting proteins are substrates of the respective SENPs is explained by the fact that anti-Flag-SENP3/SENP5 IPs enrich for larger protein complexes, where not all individual proteins are targets of SENPs. At the same time some changes in SUMOylation upon depletion of SENP3/5 could be indirect. We rationalized that screening for both interactors and targets enables us to most reliably identify direct targets by merging both datasets. We feel that this approach was highly gratifying as exemplified by the identification of PELP1 (an already known key substrate of SENP3) and UTP14A as a novel key substrate of SENP3/5. To our opinion, both approaches are very specific. In the IP approach one needs to be aware that the catalytic mutants not only trap substrates, but also proteins associated with these substrates.

It would be important to confirm sets of key substrates by IP-Westerns, not just in total lysate because there is no formal proof that the extra bands represent sumoylation.

Following the reviewer's suggestion, we have now added IP-Western experiments (new Figure 2G, Supplementary Figure 1E).

6. Figure 3c and d – Include single knockdown of SENP3 in Figure 3d and single knockdown of SENP5 in Figure 3c and d.

We agree that these controls were missing. We have now added the respective single knock-downs (now new Figure 4D, E)

7. Figure 3d – Interesting to see that knockdown of UTP14A reduces ribosome maturation. The authors should check whether sumoylation of UTP14A is important by knocking down UTP14A and rescuing with sumoylation-deficient UTP14A.

The observation that knock-down of UTP14A reduces ribosome maturation is consistent with its described function at 90S particles. We agree with the reviewer that a knock-down rescue experiments could be one approach to directly address the impact of SUMOylation. However, as outlined above due to redundancy in SUMOylation sites it is very challenging to generate SUMO-deficient UTP14A. Furthermore, as mentioned above, our data suggest that enhanced SUMOylation prevents pre-ribosome association of UTP14A making it unlikely that a SUMO-deficient mutant of UTP14A would *per se* affect ribosome biogenesis. We therefore feel that the above-mentioned endogenous IP of UTP14A in the presence or absence of SENP3/5 is a more conclusive approach to get insight how persistent UTP14A SUMOylation affects ribosome maturation.

8. Figure 4 and 5 – The authors should check whether knocking down SENP3 and SENP5 simultaneously further reduces CDK6 and strengthens the phenotype. Knockdown efficiency of SENP5 should be checked by Western.

Following the reviewer's suggestion we have now added a series of experiments, in which we co-depleted SENP3 and SENP5. In various assays co-depletion of both isopeptidases strengthens the phenotyp.

New Figure 7B and Supplementary 6C: When compared to single SENP3 and SENP5 knock-down, co-depletion of SENP3/5 aggravates the cell cycle defects in the osteosarcoma cell line U2OS (p53-proficient) or in the pancreatic cancer cell line BxPC-3 (p53-deficient).

New Figure 2G and Figure Supplementary Figure 3E: When compared to single SENP3 and SENP5 knock-down, co-depletion of SENP3/5 enhances SUMOylation of UTP14A. Accordingly, co-depletion aggravates the 40S ribosome maturation defect monitored by the Ribo-Halo assay system.

9. Figure 6 – Interesting to see that the different knockdowns reduce CDK6, but how does SUMO regulate the target proteins? It would be important to perform knockdown and rescue experiments with sumoylation-deficient mutants.

Our data indicate that the defect in ribosome biogenesis - not alterations in SUMOylation status of individual proteins - is the signal controlling CDK6 abundance through transcriptional or post-transcriptional regulation. We have now strengthened this idea by depleting a set of RPS/RPL protein (new Figure 8D). Furthermore, the concept is now also visualized in Figure 8E.

10. Do the findings have pathological relevance? The authors speculate in the discussion about the pathological relevance in AML. It would be important to provide experimental evidence for the pathological relevance of their findings.

Following the reviewer's suggestion we now experimentally addressed this issue. As

discussed in the initial version, a subset of tumour cell lines is dependent on high CDK6 expression and targeting of CDK6 is an emerging anti-cancer strategy. To explore whether SENP3/5 might be exploited to target CDK6, we used the pancreatic cancer cell line BxPC-3 as a high CDK6 model system that concomitantly exhibits low CDK4 expression and additionally lacks functional p53 (new Figure 7A). Depletion of SENP3 from BxPC-3 cells significantly reduces pRB phosphorylation again largely recapitulating CDK6 depletion (new Figure 7A). Depletion of SENP3 or SENP5 from BxPC-3 cells impairs cell cycle progression as indicated by a reduced number of cells in S phase (new Figure 7B). Co-depletion of SENP3/5 further reduces S phase cells again indicating a redundancy of both isopeptidases. To expand on this data, we performed analogous experiments in Ramos cells, a CDK6-dependent, p53-mutant B cell model. Depletion of SENP3 or SENP5 downregulates CDK6 and affects cell proliferation as indicated by a reduced number of S phase cells (Figure 7C, D). Altogether, these data indicate that SENP3/5 represents a potential vulnerability of tumors, including in p53-mutant tumours.

Reviewer #3 (Remarks to the Author):

In this Manuscript, the authors present a study on the role of nucleolar SUMO isopeptidases during ribosome synthesis. In the course of this study, the authors also discovered a new control point of altered ribosome biogenesis, independent of p53. Indeed, in the course of their work, the authors identified a role for SENP3 in the synthesis of the two ribosomal subunits and also a rather specific role for SENP5 in the assembly of the 40S subunit. Second, they identified that depletion of these factors promoted G1 arrest during the cell cycle, a known consequence of defects in ribosome synthesis. This specific response to disruption of ribosome synthesis, also known as IRBC (impaired ribosome biogenesis check-point), is mediated by ribosomal proteins released as a free pool that are able to sequester Mdm2 and promote p53 stabilization. As a marker of IRBC activation, the authors noted an increase in the expression of p53 pathway components (Figure 4). In addition to this pathway, the authors noted a clear reduction in CDK6 protein following SENP3 depletion. To test whether this reduction in CDK6 was specific to SENP3/SENP5 depletion, they also tested the effect of other ribosomal assembly factors and drug on CDK6 expression, and found that CDK6 was a novel general response independent of p53 following ribosomal stress.

All data presented are of good quality and support a function of SENP3/SENP5 in ribosome assembly as well as a specific effect of ribosome assembly defects on CDK6 expression that could explain the p53-independent response to ribosomal stress. This study is very compelling and reveals a novel p53-independent CDK6 pathway activated after ribosomal disruption, a pathway that is widely targeted by chemotherapeutic agents. This will undoubtedly benefit to the field.

Here are my comments:

1- At the beginning of the article, the authors characterize the role of SENP3 and SENP5 on the modification of two ribosome assembly factors: LAS1L and UTP14A involved in pre-60S and SSU processome processing respectively. The authors then propose that SENP3 is involved in 40S and 60S processing while SENP5 only has a function in small ribosomal subunit processing. This function is redundant with that of SENP3 in the same pathway. This is confirmed by sucrose sedimentation assays in which SENP3 co-migrates with the 40S and 60S peaks while SENP5 is mainly present in the 40S fractions. Next, to fully assess the independent role of SENP3 and SENP5, the authors set out to use a rather robust and elegant assay: the Ribo-Halo assays (Figure 3C). From my perspective, two conditions are missing in this test and could improve the outcome:

a- A check for no effect of SENP5 depletion on RPL28-halo as a negative control, to confirm that SENP5 has no role on 60S maturation and that the effect of SENP3 is not due to its role on the sumoylation of UTP14a.

b- An independent depletion assay with separate depletion of SENP3 and SENP5 on RPS3-Halo, to fully assess their independent function during small ribosomal subunit processing.

We agree with the reviewer that in the initial version of our paper specific and overlapping functions of SENP3/5 were not sufficiently addressed. To strengthen this point, we have now performed SENP5 knock-down in the Rpl28-Halo assay as suggested by the reviewer. Furthermore, we have quantified the data by quantitative image analysis. In line with our biochemical data SENP5 does not affect large subunit biogenesis (new Figure 4D). As suggested, we have performed single and co-depletion of SENP3 and SENP5 in the RPS3-Halo assays (new Figure 4D, new Supplementary Figure 3E). Further, we have quantified the data. The data demonstrate a regulatory role SENP3 - but not SENP5 - in large subunit biogenesis (new Figure 4D). By contrast both isopeptidases synergize in controlling small subunit maturation as demonstrated by impaired 40S maturation upon individual depletion of SENP3 or SENP5 (new Figure 4D) and the aggravated 40S maturation defect in co-depletion experiments (new Supplementary Figure 3E).

To more clearly dissect distinct roles of SENP3 and SENP5 in 40S/60S RiBi we now have also performed Northern Blot experiments to monitor rRNA processing. Using specific probes (within ITS1 and ITS2) that detect distinct rRNA intermediates we demonstrate that depletion of SENP3 results in the accumulation of the 32S precursor confirming our published work on the involvement of SENP3 in 32S to 28S rRNA processing (new Figure 4B and Suppl. Figure 3A). Noteworthy, depletion of SENP5 does not affect 28S maturation. However, depletion of either SENP5 or SENP3 reduces the level of the 18SE rRNA species (new Figure 4B and Suppl. Figure 3A), which represents a direct precursor of the 18S rRNA, indicating that both nucleolar isopeptidases are critical for small subunit maturation. These data are perfectly in line with the results from the HaLo assays and our biochemical assays demonstrating that SENP3 acts on both 40S and 60S trans-acting factors, while SENP5 only acts on 40S factors.

2- Regarding the effect of CDK6 loss on ribosome biogenesis, the authors suggest in their last results section and discussion that CDK6 depletion could explain part of the G1/S arrest observed in p53-deficient cells following the disruption of ribosome synthesis. To fully support this role, it would be important to overproduce CDK6 in both p53-deficient and p53-proficient cells and to analyze the effect of ribosome synthesis disruption (by CX-5461 for example) on the cell cycle. One would expect to reduce G1/S arrest in such cells compare to control cells that do not overexposes CDK6.

We now added data demonstrating that the major downstream target of our newly identified p53-independent impaired RiBi-CDK6 signaling process is indeed pRB. We show that alterations in CDK6 levels upon SENP3/5 reduction perfectly correlate with phosphorylation of pRB at S807/811 and T826 (new Figure 6J and Supplementary Figure 6D). The data demonstrate that depletion of SENP3 phenocopies CDK6 depletion (new Figure 6J, new Supplementary Figure 6D). Importantly, as suggested by the reviewer, the reduced pRB phosphorylation upon SENP3 loss could be rescued by CDK6 re-expression indicating that impaired cell cycle progression upon depletion of SENP3 is directly linked to CDK6 and alterations in the phosphorylation status of pRB (new Figure 6K, new Supplementary Figure 6E). We agree that the proposed experiment would further support this concept, but due to time constraints we were not yet able to generate a suitable CDK6-overexpressing rescue system in both p53-deficient and p53-proficient RPE1 cells.

REVIEWER COMMENTS

Reviewer #1 (Remarks to the Author):

This is a revised version of a previously submitted manuscript by Donig et al. The authors have provided a thorough point by point response to the comments and concerns that were initially raised in the first submission. I believe the manuscript has significantly improved in its cohesiveness and the new data presented help to support the conclusions drawn regarding the distinct and overlapping roles of Senp3 and Senp5 in ribosome biogenesis. Additionally, the findings that impaired ribosome biogenesis generally affects CDK6 expression, and that SENP3/5 potentially represent a therapeutic vulnerability for p53-mutant tumors are significant and could lead to the development of novel cancer strategies in the future. In conclusion, the authors have addressed most of my concerns when feasible in this revision, however, some of the new experiments presented are not sufficiently rigorous in their execution or presentation and the following point should be addressed to support the overall significance of the findings.

Comments:

Figure 3E: The authors should provide quantification of PELP1 and UTP14A in each fraction (normalized to total PELP1 or UTP14A signal from all lanes) from 3 independent experiments. Sup Figure 2A: Representative IF images should be shown in addition to the quantification graph. Please precise the number of experiments or cells counted to generate the graph.

The authors cite MS datasets identifying K733 as the major SUMOylation site in UTP14A. However, this site is not necessarily a SENP3/5 substrate. To support that SUMOylation of K733 would interfere with UTP14A incorporation into pre-ribosome, the authors should first demonstrate that K733 is a bona fide substrate by mutagenesis.

Figure 6J: The levels of p-Rb seem to follow the levels of total Rb. Is there really a reduction in Rb phosphorylation. Why are Rb levels changing? Please provide quantification of three independent experiments.

Figure 6K as presented, is not really a rescue of CDK6 re-expression, it's a supraphysiological overexpression of CDK6 that inevitably will result in phosphorylation of pRb. Endogenous CDK6 is not even visible in the siControl -Dox lane. Why are the blots cut in two? This experiment must be done from the same WB. Please show full blots with continuous signal of siC and siSENP3.

Reviewer #2 (Remarks to the Author):

My most important concerns have been properly addressed. The study has now significantly improved, especially from a molecular mechanistic point of view. I do recommend to label UTP14A in Figure 3D to highlight the equal IP efficiency.

Reviewer #3 (Remarks to the Author):

In the revised version of their manuscript, the authors have addressed all the points I raised.

Although the rescue experiment (CDK6 overexpression in p53-deficient cells) suggested earlier would have helped strengthen their result. As an alternative, they analyzed the regulation of the downstream factor pRB following CDK6 re-expression after SENP3 depletion, which is acceptable. In conclusion I have no other major comments. I think this work is a good addition to our understanding how cell react to nucleolar stress.

Point-by-point answer to reviewer comments

Reviewer #1 (Remarks to the Author):

This is a revised version of a previously submitted manuscript by Donig et al. The authors have provided a thorough point by point response to the comments and concerns that were initially raised in the first submission. I believe the manuscript has significantly improved in its cohesiveness and the new data presented help to support the conclusions drawn regarding the distinct and overlapping roles of Senp3 and Senp5 in ribosome biogenesis. Additionally, the findings that impaired ribosome biogenesis generally affects CDK6 expression, and that SENP3/5 potentially represent a therapeutic vulnerability for p53-mutant tumors are significant and could lead to the development of novel cancer strategies in the future. In conclusion, the authors have addressed most of my concerns when feasible in this revision, however, some of the new experiments presented are not sufficiently rigorous in their execution or presentation and the following point should be addressed to support the overall significance of the findings.

We were glad to hear that the reviewer acknowledges the improvements of our manuscript. At the same we took notice of his/her valuable concerns, which we addressed by including additional experimental data and/or re-evaluating previous data.

Comments:

Figure 3E: The authors should provide quantification of PELP1 and UTP14A in each fraction (normalized to total PELP1 or UTP14A signal from all lanes) from 3 independent experiments.

We thank the reviewer for this suggestion. In our revised version, we had selected an immunoblot of one representative experiment from three independent biological replicates (see below). Following the reviewer's suggestion, we now quantified the data from siControl and siSNP3/5.

To do so, we determined the signal for UTP14A in each fraction (using a defined area), across different experiments and normalized it to the input sample (representing 10% of the cleared cell extract that was loaded onto the gradient). After normalisation to the input, we performed a scaling to enable comparison of the different experiments, which each had different overall signal intensities.

In case of UTP14A, there is one clear peak of signal and this is shifted towards the top of the gradient when SENP3/5 are lacking, which is in line with the reduced signal in fractions 5-8 that we describe in the text. It is worth noting, that the main UTP14A signal is shifted to earlier fractions that do not overlap with RPS3A justifying our conclusion that a fraction of UTP14A is lost from pre-ribosomes upon depletion of SENP3/5.

In contrast to UTP14A, quantification of the PELP1 signal was rather difficult due to the spreading of the signal across many different fractions. However, by evaluating the three replicates, we still feel that we can maintain our claim that PELP1 levels are increased in the RPL3-positive fractions upon depletion of SENP3/5. We would also like to stress that in our published work we have already provided ample evidence supporting the concept that constitutive SUMOylation of PELP1 traps the complex at pre-ribosomes (Raman et al., Molecular Cell, 2016).

Replicate #2:

Replicate #3:

Sup Figure 2A: Representative IF images should be shown in addition to the quantification graph. Please precise the number of experiments or cells counted to generate the graph.

We agree that the respective IF images were missing in the initial version of the manuscript. Representative pictures are now included as Supplementary Figure 2B. In total 3 independent replicates of each sample and at least 150 cells were counted to generate the statistics (new Supplementary Figure 2C). We would like to stress that the nucleoplasmic signals of SENP5 and particularly SENP3 are weak and hard to visualize in the processed images. Microscopic evaluation as well as unbiased software-based quantitative image analysis were more conclusive.

The authors cite MS datasets identifying K733 as the major SUMOylation site in UTP14A. However, this site is not necessarily a SENP3/5 substrate. To support that SUMOylation of K733 would interfere with UTP14A incorporation into pre-ribosome, the authors should first demonstrate that K733 is a bona fide substrate by mutagenesis.

We agree with the reviewer that our conclusion of K733 being the SUMO attachment site targeted by SENP3/5 was extrapolated from published SUMO proteomics dataset. We further agree that this claim should be supported by experimental data. To this end, we performed a Ni-NTA pull down assays in control cells or cells depleted for SENP3/5. The cells were transfected with His-SUMO2 in combination with wild-type UTP14A or the respective K733R mutant. The results are shown in new Supplementary Figure 2D. As can be seen, a higher molecular anti-UTP14A reactive band (which represents the SUMOylated form) is visible only

on wild-type UTP14A, but not on the K733R variant, indicating that K733 seems to be the major SUMOylation site of UTP14A controlled by SENP3/5.

Figure 6J: The levels of p-Rb seem to follow the levels of total Rb. Is there really a reduction in Rb phosphorylation. Why are Rb levels changing? Please provide quantification of three independent experiments.

We agree with the reviewer that the quality of Figure 6J was not satisfying. We therefore performed three independent biological replicates. The data were quantified and the previous immunoblots of Figure 6J was replaced by a new version, where the ratio of pRB/RB signal is given. Quantification of two other replicates is shown below. Quantitative data from all replicates support our conclusion that depletion of SENP3 recapitulates depletion of CDK6 with respect to alteration of RB phosphorylation. As can be seen in the new datasets, these changes are not due to changes in the total level of RB.

Figure 6K as presented, is not really a rescue of CDK6 re-expression, it's a supraphysiological overexpression of CDK6 that inevitably will result in phosphorylation of pRb. Endogenous CDK6 is not even visible in the siControl -Dox lane. Why are the blots cut in two? This experiment must be done from the same WB. Please show full blots with continuous signal of siC and siSENP3.

The experiment presented in Figure 6K was indeed done from the same immunoblot. As indicated below, different concentration and duration of doxycycline treatment were tested in the same experiment. Therefore, the blot was cut, to just show relevant parts in the manuscript.

REVIEWERS' COMMENTS

Reviewer #1 (Remarks to the Author):

The authors have provided new data that addresses weaknesses that were raised in the previous review. I am satisfied with the authors responses and the new data provided is rigorous. This works is scientifically sound and provides new information that will be valuable to the fields of ribosome biogenesis and cancer. I have no further concerns.